# Subnanometre-resolution structure of the doublet microtubule reveals new classes of microtubule-associated proteins

Muneyoshi Ichikawa[1], Dinan Liu[1], Panagiotis L. Kastritis[2], Kaustuv Basu[3], Tzu Chin Hsu[1], Shunkai Yang[1] & Khanh Huy Bui[1,4]

Cilia are ubiquitous, hair-like appendages found in eukaryotic cells that carry out functions of cell motility and sensory reception. Cilia contain an intriguing cytoskeletal structure, termed the axoneme that consists of nine doublet microtubules radially interlinked and longitudinally organized in multiple specific repeat units. Little is known, however, about how the axoneme allows cilia to be both actively bendable and sturdy or how it is assembled. To answer these questions, we used cryo-electron microscopy to structurally analyse several of the repeating units of the doublet at sub-nanometre resolution. This structural detail enables us to unambiguously assign α- and β-tubulins in the doublet microtubule lattice. Our study demonstrates the existence of an inner sheath composed of different kinds of microtubule inner proteins inside the doublet that likely stabilizes the structure and facilitates the specific building of the B-tubule.

[1] Department of Anatomy and Cell Biology, McGill University, Montréal, Québec, Canada H3A 0C7. [2] Structural and Computational Biology Unit, European Molecular Biology Laboratory, Heidelberg 69117, Germany. [3] Facility for Electron Microscopy Research, McGill University, Montréal, Québec, Canada H3A 0C7. [4] Groupe de Recherche Axé sur la Structure des Protéines (GRASP), Montréal, Québec, Canada H3G 0B1. Correspondence and requests for materials should be addressed to K.H.B. (email: huy.bui@mcgill.ca).

Cilia and flagella are organelles responsible for cell motility and sensory function[1]. Motile cilia oscillate to propel cells or mediate the movement of extracellular fluids. This motility originates from the bending movement of the cilia caused by power strokes of the axonemal dyneins anchoring within those cilia. Non-motile cilia, called primary cilia, are found in nearly every cell in the body and play important roles in mechanical and chemical sensory functions. For instance, primary cilia on the dendritic knob of the olfactory neuron are important for chemo-sensing while cilia in the nervous system function as mechano-sensors of the cerebral-spinal fluid[1]. Owing to their diverse functions, defects in ciliary components and assembly may lead to malfunctions known as ciliopathies, namely cilia-related diseases[2].

Cilia share a canonical architecture composed of nine outer doublet microtubules (doublets), either surrounding two central singlet microtubules in the case of motile cilia (9 + 2), or without a central singlet for non-motile cilia (9 + 0). Multiple copies of about 500 different proteins are needed to build a cilium[3]. Many ciliary proteins are organized into functionally distinct subcomplexes, such as dynein arms, radial spokes and nexin-dynein regulatory complexes that attach to the doublet periodically[4–9].

Conserved in both motile and primary cilia, the doublet is the elementary cytoskeleton of the cilium, comprising α- and β-tubulin heterodimers, forming a complete 13-protofilaments (PFs) A-tubule and an incomplete 10-PFs B-tubule[10,11]. While the lateral interaction between PFs in in vitro reconstituted singlet microtubules is well-understood[12], knowledge of the lateral interaction of the doublet and at the outer junction where the B-tubule is built upon the A-tubule is very limited. The doublet is very stable and robust. It does not break under significant stress during rapid ciliary/flagellar beating with a frequency range of between ∼16 to 70 Hz[13,14]. The outstanding stability of the doublet is likely linked to microtubule inner proteins (MIPs), which bind firmly to the inner wall of the doublet microtubule[10,11,15]. The densities of MIPs have been shown to be mostly similar across several species[7,11]. MIP densities show either 16- or 48-nm periodicity. Although MIPs are expected to be essential for the stability of the doublet, there is limited information about the identities and functions of the MIPs.

The axoneme and the doublet are currently resolved only to low resolution (∼20–40 Å)[5,6,11,15], which does not allow insights into the biophysical properties of the doublet and the roles of MIPs in stabilizing and forming periodic units on the doublet. Low resolution hinders our understanding of the molecular architecture and interactions of tubulins within the lattice and the MIPs with tubulins. At sub-nanometre resolution, protein secondary structures can be visualized, enabling the unambiguous fitting of atomic structures. Consequently, this can lead to the understanding of the molecular interactions within the tubulin lattice and how the MIPs contribute to the stability and periodicity of the axoneme.

In this study, we obtain high-resolution structures of multiple repeat units of the doublet by single particle cryo-electron microscopy (cryo-EM), reveal the remarkable interactions at the outer junction, and discover an inner sheath of MIPs inside the doublet. The complex and unique architecture of the inner sheath suggests that it might contribute significantly to both stability and assembly of the doublet.

## Results

**Doublet microtubule tubulin lattice architecture by cryo-EM.** To structurally analyse the doublet, cilia were isolated from *Tetrahymena thermophila* and split into individual doublets using ATP treatment. To reduce sample heterogeneity, dynein arms and radial spokes were removed by high salt treatment and dialysis before vitrifying for cryo-EM (Supplementary Fig. 1a). Micrographs of the vitrified doublets were acquired (Supplementary Fig. 1b) and the density map of the doublet was reconstructed from the 8-nm non-overlapping segments by single particle analysis at 5.7 Å overall resolution (Fig. 1a and Supplementary Fig. 1c). Local resolution analysis shows that, the A-tubule was resolved better compared to the B-tubule, especially at the ribbon region (the junction site of the A- and B-tubules) (Supplementary Fig. 1d), reflecting the high stability of this region.

Our high-resolution map allowed the assignment of α- and β-tubulins unambiguously in the doublet microtubule lattice by both visual inspection and cross-correlation (Methods section and Supplementary Fig. 1e). However, the resolution was limited for accurate modelling of the tubulin dimer. To obtain a higher resolution structure of tubulin for modelling, we generated the PF map by boxing out each PF from the doublet density map, aligning and averaging. This led to the map of the PF at 4.6 Å resolution (Fig. 1b and Supplementary Fig. 1c). *T. thermophila* tubulin dimer structure was modelled based on previous tubulin structure[16] and refined using PHENIX.refine[17] in this PF map (Fig. 1b). The above tubulin model was then fitted to the doublet density map to construct a pseudo-atomic model of the doublet (Fig. 1c). Tubulin model structures fitted well in the doublet map except for PF-B1 (discussed later).

The α- and β-tubulin assignment is consistent with a cryo-electron tomography study of the doublet partially decorated with kinesins[15]. As previously suggested, both the A- and B-tubules have B-lattice staggers[15]. The microtubule lattice seam in which α- and β-tubulin subunits interact laterally is identified between PFs-A9 and A10 (Fig. 1c, red arrow).

Next, we quantified the local curvature of the doublet lattice. Local curvature of the microtubule lattice is in an inverse relationship with the rotation angle between neighbouring tubulins since the H1'-S2 and H2-S3 loops and the M-loop of the adjacent tubulin mediate the lateral interaction[16,18] and act as the hinge between neighbouring tubulins (Supplementary Fig. 2a). Therefore, we measured the rotation angles between fitted successive PF models and compared them to singlet microtubules with different PF numbers. The results suggest that the local curvatures of PFs in the A- and B-tubules are highly heterogeneous (Supplementary Table 1 and Supplementary Fig. 2b). The A-tubule, despite being composed of 13 PFs, displays a wide range of local curvatures from 9-PF-like to 22-PF-like microtubules. The different local curvatures lead to the observed distortion in the A-tubule as previously reported[10,15]. Regarding the B-tubule, in contrast to the prevailing assumption that it has a uniform 15-PF-like global curvature[10], there are not only 15-PF-like PF pairs but also 13-, 16-, 17- and 18-PF-like curvatures.

**Outer junction structure.** At the outer junction, where the tubulin from the B-tubule attaches to the tubulin from the A-tubule, the M-loop regions of our fitted tubulin models in the PF-B1 stick out from the density map (Supplementary Fig. 2d). The M-loop of α-tubulin seems to adopt a different conformation while there is no observed density for the M-loop of β-tubulin. To account for the specific conformation of PF-B1, we performed further real space refinement of our tubulin model in the PF-B1 density (Supplementary Fig. 2d).

While interactions between tubulins in PFs-A10 and A11 are canonical lateral interactions mediated by the H1'-S2 and H2-S3 loops of tubulin in PF-A10 and the M-loop of the tubulin in PF-A11 (refs 16,18), the interactions between PF pairs-A10/B1 and A11/B1 are specific to the doublet (Fig. 2a). To evaluate the

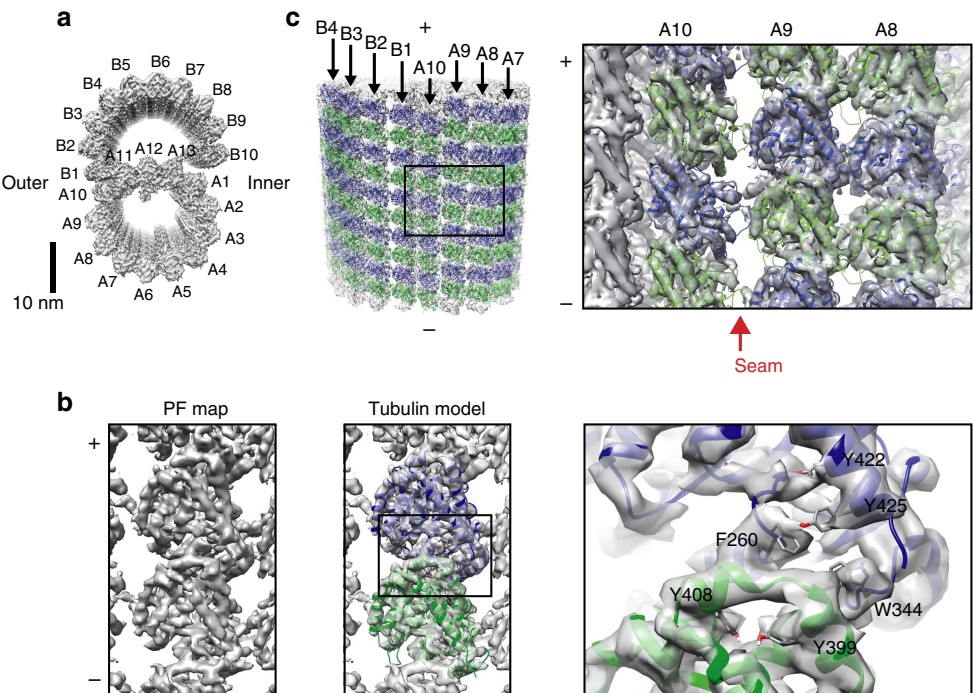

**Figure 1 | The tubulin lattice of the doublet microtubule.** (**a**) The surface rendering of the 8-nm averaged doublet density map viewed from the tip of the cilia (plus end of microtubule). PF numbering, the outer and inner junction sides are indicated. (**b**) The surface rendering of the PF density map viewed from the outside (left panel). The refined model of *Tetrahymena* tubulin dimer fitted in PF map (α-tubulin: green; β-tubulin: blue; middle panel). Magnified view of the tubulin model fitted inside the PF map shows visible bulky side chains in the density map (right panel). (**c**) Pseudo-atomic model of the doublet viewed from the outer junction side (left panel) and the magnified view of PFs-A8-10 region (right panel). Red arrow indicates the seam location. (+) and (−) signs indicate plus and minus-ends of microtubule.

strength of interactions at the outer junction, we performed energy calculations of the interfaces among PFs-A10, A11 and B1 (Supplementary Table 2). Our calculations show that the interfaces between pairs of α- and β- tubulins from PFs-A10 and B1 (Fig. 2b) are relatively large (buried surface area of 688.6 Å$^2$ and 462.1 Å$^2$ for α- and β-tubulins, respectively). The interface between α- and β-tubulins from PFs-A11 and B1 (Fig. 2e) are weaker with buried surface area of 351.7 and 137.5 Å$^2$ for α- and β-tubulins, respectively. Although each number appear not that high, the strength of interactions in the outer junction will multiply as more tubulins are incorporated into PF-B1.

Using PDBePISA, an interface prediction server[19], these interactions are predicted to be mediated by electrostatic interactions involving residues R308 of α-tubulin of PF-B1 to E414 and E417 of PF-A10 and residue R306 of β-tubulin of PF-B1 to E410 of PF-A10 (Fig. 2c,d). The M-loop of α-tubulin of PF-B1 is also predicted to interact with α-tubulin of PF-A11 through a salt bridge (H283 of PF-B1 to D306 of PF-A11; Fig. 2f). The candidate residues involved in this interaction were highly conserved in tubulins from many species (Supplementary Fig. 2c), suggesting that these residues are important. However, these interacting residues are predictions and must be verified experimentally in future study.

**Inner junction structure.** Previously, a non-tubulin continuous density was found between PFs-A1 and B10 at the inner junction[7,11]. FAP20 was proposed to be a candidate for the non-tubulin continuous density by cryo-electron tomography combined with protein tagging[9]. In our structure, we were not able to observe this density (Fig. 1a). The anchoring of the

B-tubule to the A-tubule at the inner junction is mediated by MIP densities inside the B-tubule (see later section). There is a possibility that most of FAP20 have dissociated from the doublet after NaCl extraction and dialysis. Therefore, the inner junction non-tubulin structure might not be essential for the intactness of the B-tubule, consistent with previous study showing that doublets can be formed without the inner junction structure[9].

**Microtubule inner proteins.** MIPs have been shown to have 48- and 16-nm periodicities[11,15], and therefore, we reconstructed the above data with 48- and 16-nm periodicities and obtained the density maps with overall resolution of 8.6 and 6.2 Å, respectively (Supplementary Fig. 1c). The local resolutions of the MIPs seem to vary from the overall resolution, probably due to flexibility (Supplementary Fig. 1d). To better characterize the morphology of the MIPs, we subtracted the density maps with a simulated density map of the tubulin lattice (Supplementary Fig. 3a). In the doublet map with 48-nm periodicity, we observed MIP1–MIP6 at the same positions as identified in Maheshwari *et al.*,[15] (Fig. 3) along with other densities (Supplementary Notes 1 and 2).

**Filamentous MIPs, a new class of microtubule inner proteins.** Our sub-nanometre resolution maps enabled us to visualize a new class of MIPs. Surprisingly, besides the globular densities of MIPs characterized previously[15], we observed as many as 11 filamentous structures running in-between PF pairs. We named them as filamentous MIPs (fMIPs). There were four fMIPs in the A-tubule (between PFs-A6A7; A7A8; A11A12; and A12A13) and seven fMIPs in the B-tubule (between PFs-B2B3;

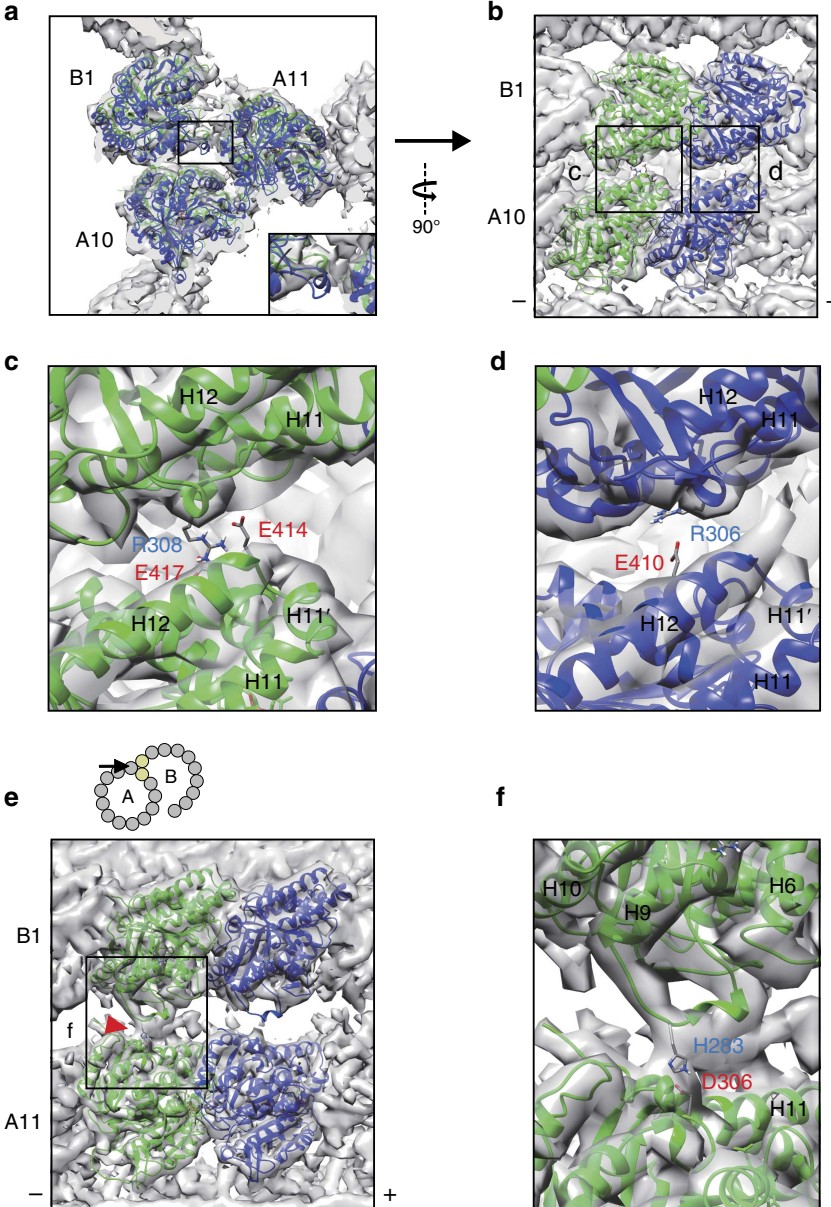

**Figure 2 | Structure of the outer junction.** (**a**) The outer junction region highlighted with fitted tubulin molecules in PFs-A10, A11 and B1 viewed from the tip of cilia. Tubulin model fitted in PF-B1 is refined in PF-B1 density map. Inset shows the M-loop of β-tubulin in PF-B1 is sticking out from the density map. (**b**) Perpendicular view of the interactions between PFs-A10 and B1. (**c,d**) Magnified views (black boxes in **b**) show the residues predicted to be involved in non-canonical tubulin–tubulin interactions between tubulins of PFs-A10 and B1. (**e**) View of interactions between PFs-A11 and B1. Angle and location of the view is indicated in the cartoon. Red arrowhead shows the interaction site between the M-loop of α-tubulin of PF-B1 and α-tubulin of PF-A11. PF-A10 is removed in this view for visualization. (**f**) Magnified view from **e** showing the residues predicted to be involved in the PFs-A11 and B1 interaction. ( + ) and ( − ) signs indicate plus and minus-ends of microtubule. The α-helix numbers are indicated in **c,d** and **f**.

B3B4; B4B5; B5B6; B6B7; B7B8; and B8B9; Fig. 3, red arrows). fMIPs are found in the furrows between PFs that are devoid of globular MIPs (Supplementary Table 3); in other words, all PF pairs were connected by protein densities. Based on its thickness, fMIPs are likely to be only single α-helices stretch (Supplementary Fig. 4d).

Even though fMIPs are found mostly in similar places relative to the tubulin PFs (Supplementary Fig. 4c), there are certain differences (Figs 3b–f and 4 and Supplementary Fig. 4a and Supplementary Table 4). Some fMIPs are straight while others are curved. fMIP-A6A7 is mostly straight, but there is an abrupt turn at one end that extends towards PF-A6 and reaches to the base of MIP1a (Figs 3b and 4a, white arrowhead). Both fMIPs

in the ribbon region (fMIPs-A11A12 and A12A13) are straight (Fig. 4c,d). In contrast to fMIP-A12A13 with no apparent discontinuity, fMIP-A11A12 has two longitudinal discontinuities (Fig. 4c, red arrowheads). Since these features are visible only in 48-nm map, it is highly likely that these fMIPs are in 48-nm repeating unit except for fMIP-A12A13, which appears similar in both 48-nm and 16-nm maps (Supplementary Fig. 4b). However, it is also possible that some fMIPs might have 96-nm periodicity.

The fMIPs seem to interact with tubulins along its length (Fig. 5a,b and Supplementary Fig. 4c). Bifurcations from fMIP-A12A13 stick into a hole located at the centre of four neighbouring tubulin dimers every 16 nm (Fig. 5a,b, red

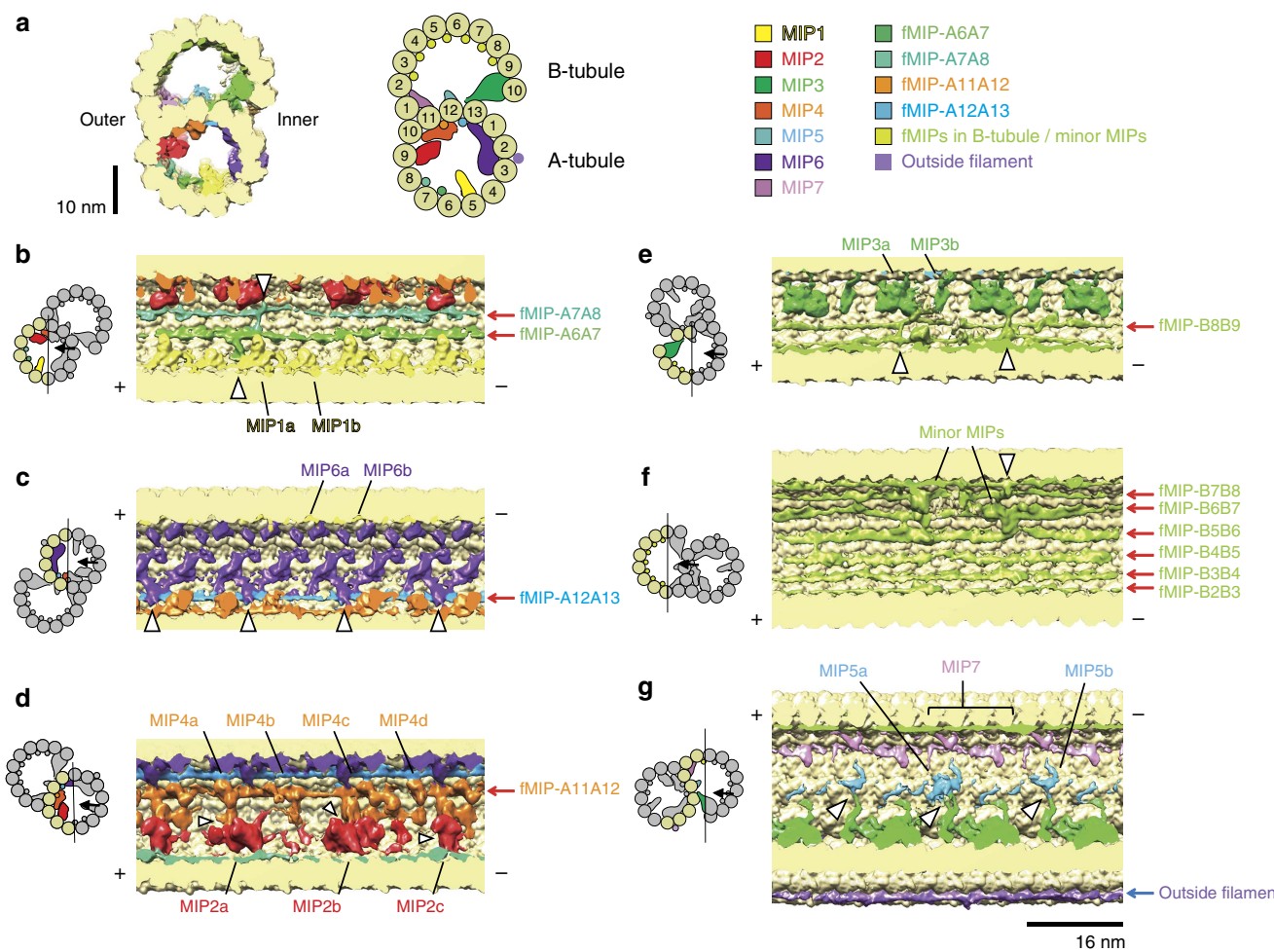

**Figure 3 | Structures and locations of MIPs.** (**a**) Cross section of the surface rendering of 48-nm repeat doublet density map (left) and corresponding cartoon (right) with the microtubule associated protein coloured. Our nomenclature of globular MIPs is adapted from Maheshwari et al.,[15].
(**b–g**) Longitudinal cross sections of the surface rendering of the 48-nm map focusing on MIP1 and fMIPs-A6A7 and A7A8 in **b**, MIP6 and fMIP-A12A13 in **c**, MIP2/4 and fMIP-A11A12 in **d**, MIP3 and fMIP-B8B9 in **e**, the other fMIPs and minor MIPs (Supplementary Note 1) in the B-tubule in **f**, and MIP5 and MIP7 in **g**. In **b–g** tip of the cilia (+ end) is toward the left side. Black lines and arrows in the cartoons of the doublet on the left indicate the sections and viewing directions on the right. Red arrows from the side represent fMIPs inside the doublet and blue arrow indicates continuous density binding outside the doublet. White arrowheads show lateral contacts between neighbouring MIPs. Unsharpened map was used for the generation of figure. The colours of the MIPs are consistent in all figures.

arrowheads). The other fMIPs in the A-tubule and some fMIPs in the B-tubule also show insertions into the tubulin lattice (Supplementary Table 4). Using the well-resolved fMIP-A12A13 in 16-nm averaged map as a model, we identified four regions in α- or β-tubulins that are likely to be important for lateral interaction with fMIP-A12A13 (Fig. 5c and Supplementary Fig 4e). While T56-K60 (TGAGK) and S277-Q285 (SAEKAYHEQ) from α-tubulin and 54A–58R (ATGGR) and R276-283A (RGSQQYRA) from β-tubulin are also involved in the canonical lateral contact of PFs, the other regions P32-T41 (PDGQMPSDKT) and V362-V372 (VVPGGDLAKV) from α-tubulin and D31-G38 (DPTGTYHG) and P358-K362 (PKGLK) from β-tubulin have not been described to be involved in PF lateral interaction. We performed multi-sequence alignment of α- and β-tubulins from various organisms and examined the sequence conservation of the four identified regions above. Those regions are highly conserved in ciliated organisms and, in contrast, show variation in non-ciliated organisms (Fig. 5d and Supplementary Fig. 4f). This suggests that fMIP interaction is highly and specifically important for the architecture of the doublet.

**Intra- and inter-interactions between MIPs and tubulins**. In addition to the finding of fMIPs, our map provides more significant details on the previously characterized globular MIPs[7,11,15] (Figs 3, 6 and 7 and Supplementary Fig. 3). We also observed a 16-nm repeating unit density connecting the outside of PF-A11 to the inside of PF-B1 and named sequentially as MIP7 (Figs 3g and 6i,j), which is previously visualized as a laminar sheet[15]. MIP7 interacts with PFs-A11, B1 and B2 (Fig. 6i,j) and probably acts as a structural bridge between the A- and B-tubules at the outer junction region.

The globular MIPs show various interactions with the tubulin lattice (Figs 6 and 7 and Supplementary Fig. 3b–d). We observed densities from globular MIPs protruding into the tubulin lattice in different patterns (Figs 6 and 7, red arrowheads). For example, the basal regions of MIP2 densities stick into the seam between PFs-A9 and A10 (Fig. 7c,d), while the basal regions of MIP1 densities stick into the furrows between two tubulin dimers and between α- and β-tubulins of the same dimers along PF-A5 (Fig. 6a,b). Since almost the entire inner surface of the doublet is covered with protein densities, we visualized the sequence conservation of the entire tubulin dimer

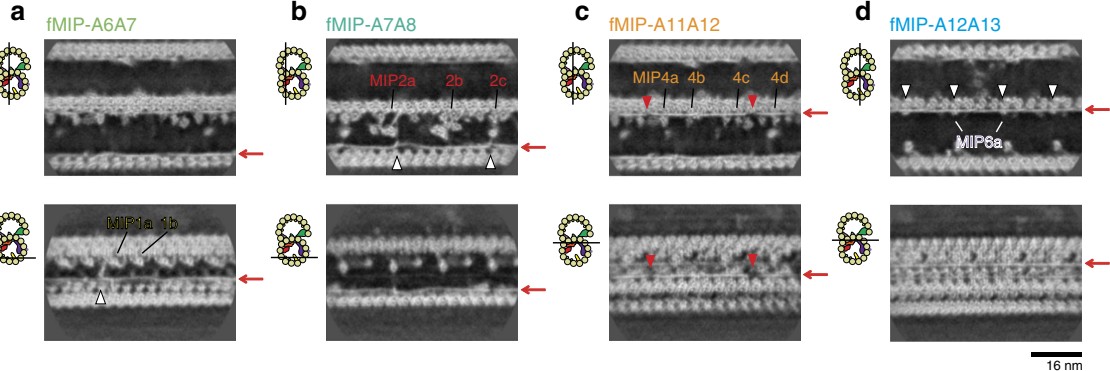

**Figure 4 | Morphology of fMIPs and their interactions with globular MIPs.** Longitudinal sections of the 48-nm averaged density map showing fMIPs in the A-tubule focusing on fMIP-A6A7 (**a**), fMIP-A7A8 (**b**), fMIP-A11A12 (**c**) and fMIP-A12A13 (**d**). Locations of fMIPs are indicated by red arrows. Different types of morphology such as curves, discontinuities (red arrowheads) and interactions with globular MIPs (white arrowheads) are seen. Sections are indicated by black lines in the cartoons on the left side.

(Supplementary Fig. 5). Consistent with our fMIP interaction analysis, loops and α-helices (H1, H2, H6, H7, H9 and H10) in the lumen side of the tubulins from ciliated organisms are highly conserved compared to the non-ciliated species. For tubulins from non-ciliated species, the inside surface is less conserved even compared to its outside surface, suggesting few proteins are bound inside the cytoplasmic microtubule *in vivo*.

MIPs do not just bind along the tubulin lattice, but also interact with other MIPs laterally. There are four types of lateral contacts including (i) globular MIPs with fMIPs, (ii) fMIPs with fMIPs, (iii) globular MIPs with globular MIPs laterally and (iv) from inside to outside of the tubulin lattice.

For type (i) interaction, MIP4 is the only MIP that binds to the top of fMIP (Figs 3d, 4c and 7e). MIP2a and 2c interact with fMIP-A7A8 (Fig. 4b, white arrowheads). MIP6a, but not 6b, has two finger-like structures holding fMIP-A12A13 (Fig. 6h, white arrowheads). Regarding the interaction of fMIP with fMIP, fMIP-A7A8 and fMIP-A6A7 were connected by a lateral filament (Fig. 3b, upper white arrowhead). The B-tubule also has two lateral filaments starting from MIP3a (Fig. 3e, white arrowheads). One of the lateral filaments of the B-tubule crosses until fMIP-B7B8, and the other lateral filament continues to fMIP-B5B6 (Fig. 3e,f, white arrowheads). These lateral contacts possibly stabilize the B-tubule. An example of the type (iii) interaction is the pairwise interactions between MIP2a and MIP4a, MIP2b and MIP4c, and MIP2c and MIP4d (Fig. 7e, white arrowheads).

The most remarkable interaction that we observed was the type (iv) in which MIPs interact across the tubulin lattice. The branches from MIP2a and MIP4c go through the A-tubule lattice hole located in the middle of the four tubulin dimers from PFs-A10 and A11 and connect to MIP7 (Fig. 7b,f,g blue arrowheads). Such interactions between proteins from inside and outside of the microtubule surface have not been characterized before.

## Discussion
The high-resolution cryo-EM maps of the doublet enabled us to visualize the doublet microtubule lattice and the inner sheath formed by fMIPs and globular MIPs inside the doublet. While *in vitro* reconstituted singlet microtubules have been solved to near-atomic resolution[16], our cryo-EM map represents *ex vivo* microtubule-based structure at sub-nanometre resolution. By assigning α- and β-tubulins in the doublet completely, we revealed the location of the seam, the complex local curvature, the

non-canonical tubulin–tubulin interaction at the outer junction, and the interactions between MIPs and the tubulin lattice.

At the outer junction, the B-tubule stably binds to the outside of the A-tubule. PF-B1 shows a non-canonical interaction with PFs-A10 and A11. The tubulin interface between the PF-B1 and the A-tubule is calculated to be strong enough to sustain the attachment of the B-tubule. This is consistent with an earlier study which showed that with over-supplied tubulin dimers, a hook structure like the B-tubule could be built in other regions of the A-tubule and also from cytoplasmic microtubules[20].

How does tubulin in PF-B1 bind specifically between PFs-A10 and A11? One possibility is that the different local angles of PF pairs in the A-tubule (Supplementary Fig. 2b) facilitate that specificity. Since tubulin from the PF-B1 is interacting with both tubulin molecules from PFs-A10 and A11 simultaneously, there might be a favourable angle of PFs for the formation of the outer junction. This is the reminiscence of microtubule-associated proteins-like EB1 and doublecortin which prefer to bind to the 13-PF microtubule[21,22]. If this hypothesis is correct, then the C-tubule in the centriole presumably cannot have the same non-canonical interaction with the B-tubule because of the small curvature of the B-tubule. Consistent with this idea, a previous centriole map from *Chlamydomonas* showed the attachment of the C-tubule to the B-tubule at the outer junction side is solely mediated by non-tubulin densities[23] (Supplementary Fig. 2e). To further test this idea, hypothetical outer junction structures are generated using each PF pair from the A-tubule (Supplementary Fig. 2f). At PF pairs with extremely low curvature like PF pairs-A11/A12 and A12/A13, steric clash occurred. At PF pair-A9/A10 with high curvature, salt bridges were not formed because of large gap. Thus, the existence of different local curvatures probably narrows down the possible binding sites of tubulin in the A-tubule. The binding of tubulin might be further facilitated and stabilized by MIP7, which tethers to both tubulins in PFs-B1 and A11. The binding of MIP7 probably makes the doublet stable in our high salt treatment.

PF-B1 is a unique region in the doublet since this is the only PF binding to the outside surface of the A-tubule. From our fitting, we found that the M-loops of tubulins from PF-B1 show different conformations from those from the other PFs. We characterized a conformational change of the M-loop of α-tubulin (Fig. 2e and Supplementary Fig. 2d). Such conformational changes have not been seen in tubulin structures in singlet microtubules[16,18]. We also detected salt bridges between α-tubulins from PFs-B1 and A11 after the refinement of the

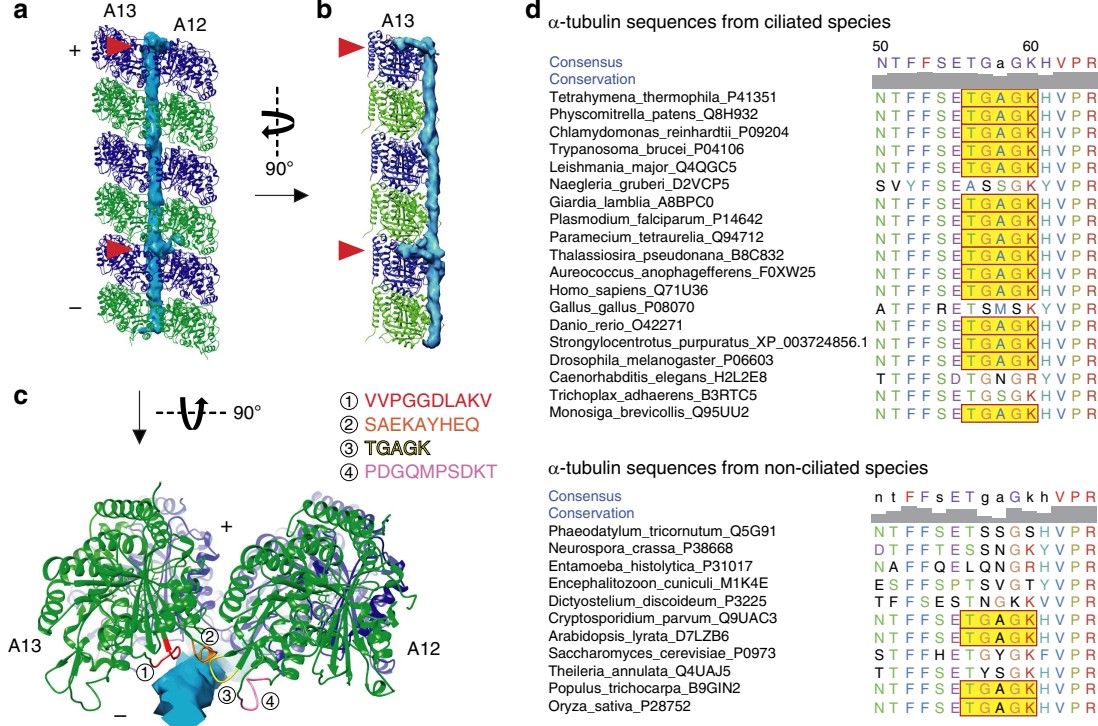

**Figure 5 | Interactions between fMIP and the tubulin lattice.** (**a**–**c**) Views of the surface rendering of fMIP-A12A13 from the 16-nm averaged map together with models of PFs-A12 and A13 from outside the A-tubule (**a**), from the side of PF-A13 (**b**) and along the PF axis (**c**). Insertions to the tubulin lattice are indicated by red arrowheads in **a** and **b**. Four possible regions of α-tubulins from PFs-A12 and A13 interacting with fMIP-A12A13 is shown in **c**. (+) and (−) signs indicate plus and minus-end of microtubule. (**d**) The multiple sequence alignment of TGAGK region showing high conservation in the ciliated organism but not in the non-ciliated organism (TGAGK is highlighted in yellow). Residue numbers of *Tetrahymena* tubulin are indicated above. Sequences of other regions and the β-tubulins are shown in Supplementary Fig. 4.

tubulin model in the PF-B1 density (Fig. 2f). This conformational change in the M-loop of α-tubulin of PF-B1 is likely to be induced by the interactions with the PF-A11. In contrast, there is no density for the M-loop of β-tubulin of PF-B1. The M-loop of β-tubulin is known to be disordered by itself and forms secondary structures when it is interacting with neighbouring tubulin molecule[24]. Therefore, the M-loop region of β-tubulin of PF-B1 might not be involved in interaction with the PF-A11 and adopt a flexible conformation. This is consistent with our energy calculation result that the interaction between β-tubulins of PFs-A11 and B1 is the lowest (Supplementary Table 2).

Here, we characterized fMIPs in between the PFs. These fMIPs are thought to stabilize the PF interactions since they are binding at the interface of the adjacent PFs (Fig. 5 and Supplementary Fig. 4c). There are two filamentous proteins known to be associated with the doublet, most likely at the PF ribbon region: tektin[25] and Rib43a (ref. 26). fMIPs observed at the ribbon of our *Tetrahymena* doublet is more likely to be Rib43a from the following reasons. First, these fMIPs are too thin for a tektin filament which is known to be ∼5 nm in diameter[25] and rather similar with single α-helix like Rib43a (ref. 26; Supplementary Fig. 4d). Second, Yanagisawa *et al.*[9] recently proposed that in *Chlamydomonas*, tektin forms the inner junction non-tubulin structure together with FAP20 and PACRG. In *Tetrahymena*, there are two predicted isoforms of Rib43a at 142 aa and 280 aa length. Our mass spectrometry analysis detected both Rib43a homologues in our doublet sample. Therefore, it is possible that either fMIP-A11A12 or A12A13 is formed by Rib43a. For the identities of the other fMIPs in the A- and B-tubules, structural analysis aided by protein tagging in the future study will reveal this point.

The inside of the doublet is almost fully decorated with MIPs and displays a complex network of longitudinal and lateral interactions (Fig. 7a and Supplementary Tables 3 and 4). The MIPs either bind to the inside surface of tubulins, such as globular MIPs, or between the PFs like fMIPs (Supplementary Tables 3 and 4). Some of the MIPs also form lateral contacts across several PFs and/or with different MIPs (Figs 6 and 7). It is noteworthy that MIPs in the A-tubule are all connected and MIPs are forming an inner sheath inside the tubulin lattice.

There can be several consequences of having an inner sheath of MIPs inside the doublet. First, the inner sheath of MIPs provides extra stability for the doublet. This is likely to prevent dynamic instability and catastrophe, which is certainly suitable for the role of cilia as the organelle responsible for cell motility. In our experience, we rarely observed PFs peeling out of the broken doublet after sonication, unlike the tip of the normal cytoplasmic microtubule[27]. Since the doublet experiences constant distortion under high curvature and shear force during ciliary bending, the presence of the MIP inner sheath might help to prolong the failure threshold for the lateral interaction of the PFs under these conditions[28]. It is known that microtubule-associated proteins (MAPs), such as MAP2, MAP4, and tau, bind to the outside of the cytoplasmic microtubule and stabilize the microtubule lattice[29]. It is interesting to see the different biophysical effects of inside and outside binding of proteins to microtubules.

Second, lateral contacts might play a scaffolding role. In *Chlamydomonas*, assembly of the beak structure was shown to be dependent on the inner junction structure despite these two structures being located several PFs away in the B-tubule[9]. This suggests some scaffold MIPs are essential for the assembly of the doublet.

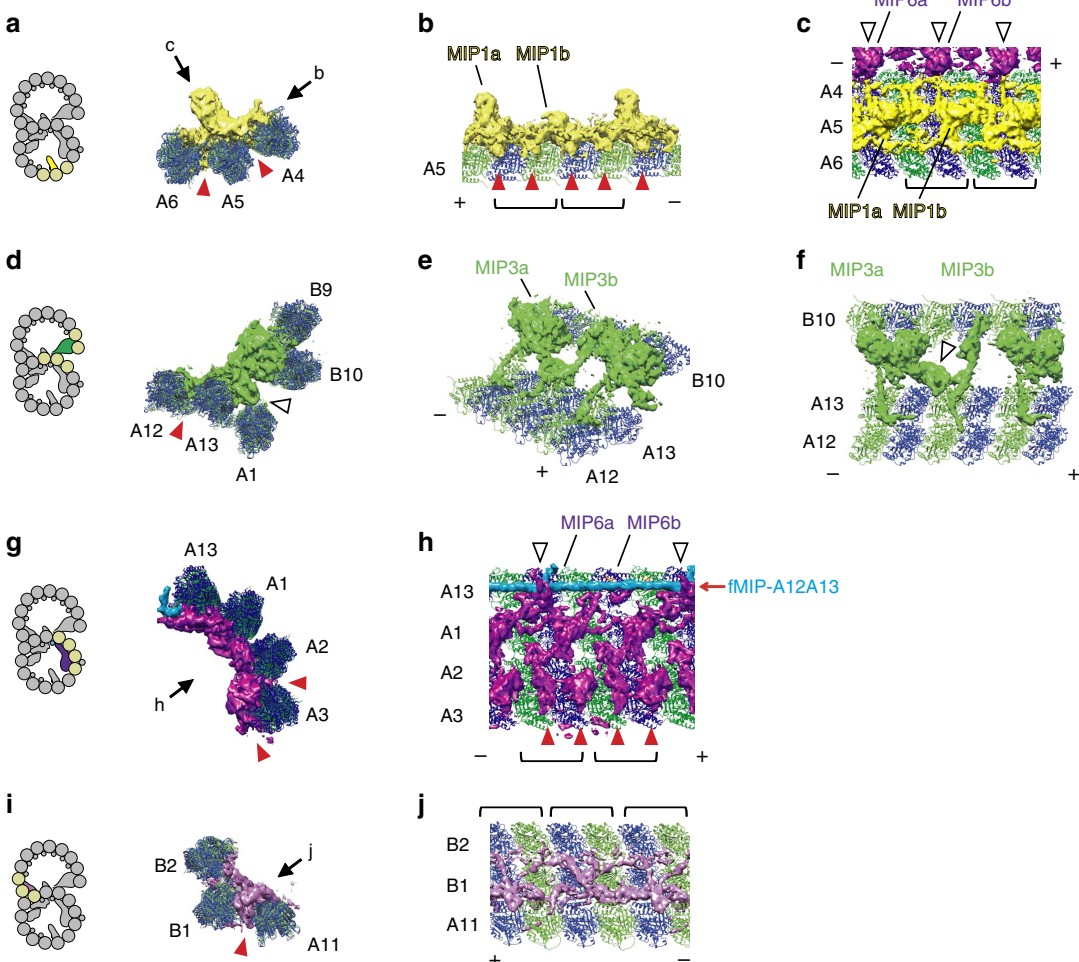

**Figure 6 | Detailed structure of MIPs.** (**a–c**) Views showing interactions between MIP1 and tubulin lattice. MIP1 consists of MIP1a and MIP1b, which are 8-nm longitudinal apart. MIP1 mainly binds on PF-A5 and connects longitudinally to each other. Red arrowheads indicate the base part of MIP1 sticking into the tubulin lattice. The part of MIP6 is shown in **c** to show the interactions between MIP1 and MIP6 (indicated by white arrowheads). (**d–f**) Views of interactions between MIP3 and the tubulin lattice. MIP3 including MIP3a and 3b densities interacts with the outside of the A-tubule (PFs-A12 and A13) and inside lumen of the B-tubule (PFs-B9 and B10) simultaneously. Red arrowhead in **d** indicates the MIP3 region binding between the PFs-A12 and A13. Longitudinal interactions between MIP3a and 3b are indicated by white arrowheads in **d** and **f**. (**g,h**) Views showing interactions between MIP6a and fMIP-A12A13 (white arrowheads) together with PFs-A1 to A3, and A13. MIP6 appears to have in fact a 16-nm repeating unit instead of 8 nm as previously reported[15] and is composed of MIP6a and 6b. Both MIP6a and 6b have at least four subdomains and were interacting intensively with PFs-A1, A2, A3 and A13. Red arrowheads indicate the densities from MIP6 protruding into the tubulin lattice. (**i,j**) Views showing MIP7 binding on the tubulin lattice. Insertion of MIP7 is shown by red arrowheads in **i**. PFs-A11, B1 and B2 are connected by MIP7. All views in **a,d,g** and **i** are from the tip of cilia (+ end). (+) and (−) signs indicate plus and minus-end of microtubule.and tubulin dimers are indicated by brackets. 16-nm structure was used in this figure and the colouring is consistent with Fig. 3.

Finally, the binding of MIPs inside the doublet lumen might induce the conformational change of PFs, which facilitate the specific assembly of the doublet such as the outer junction formation (discussed below). The fMIPs between almost every pair of PFs in the B-tubule potentially lock the PF pairs of the B-tubule in their specific curvatures (Supplementary Table 1). Unlike the 15- or 16-PF *in vitro* reconstituted singlet microtubule, the PFs in the B-tubule are straight, possibly allowing dynein-2 and kinesin-2 to drag the intraflagellar transport trains all the way along both the A- and B-tubules simultaneously without changing their tracks and colliding[30].

Currently, not many proteins are known to tightly bind to the inside surfaces of tubulin. α-tubulin acetyltransferase, also called MEC-17, was shown to bind inside the cytoplasmic microtubules[31]. However, the α-tubulin acetyltransferase does not bind stably to the microtubule. Thus, the binding patterns of MIPs/fMIPs revealed by our results are different from those previously characterized microtubule-binding proteins.

The MIPs interact exclusively with the inner surface of α- and β-tubulins. Our sequence analysis of the potential interaction regions from α- and β-tubulins from many ciliated and non-ciliated organisms remarkably shows that the MIP interactions put an evolutionary pressure on the tubulin in ciliated organisms, making these interaction surfaces more conserved. It may be possible to learn more about the conserved interaction interface between MIPs and the tubulins if we can obtain the near-atomic structure of the doublet and identify and localize various MIPs.

On the basis of our structural analysis, the possible scheme of doublet formation is presented in Fig. 8. The A-tubule is the first one to be assembled likely simultaneously with the binding of globular MIPs/fMIPs (Fig. 8b, (i)). The binding of the

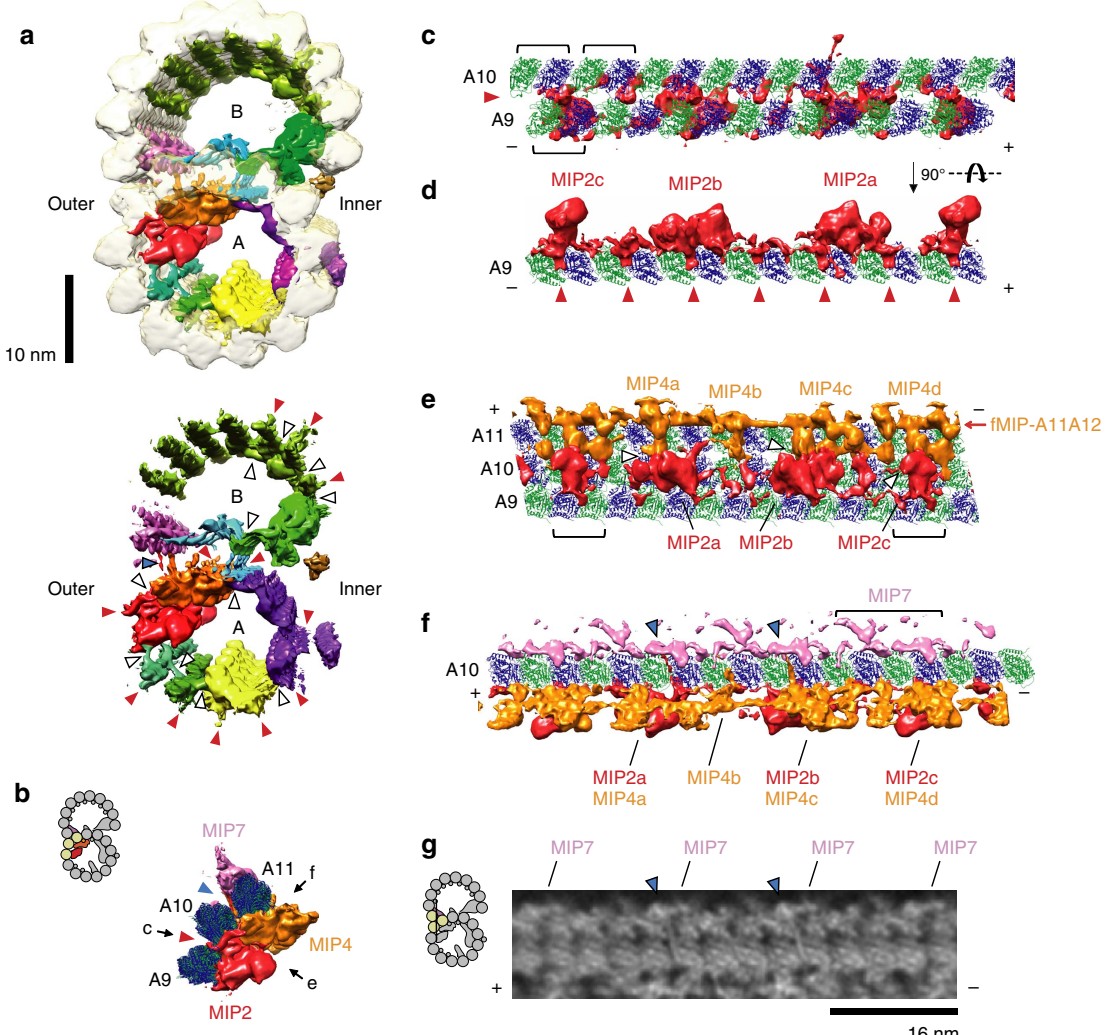

**Figure 7 | Interactions among MIPs. (a)** Difference map between our doublet map with (upper row) and without (lower row) translucent tubulin lattice viewed from tip of the cilia ( + end). **(b)** MIP2, MIP4 and MIP7 with PFs-A9 to A11 viewed from tip of the cilia ( + end). **(c,d)** Views of MIP2 interactions with the tubulin lattice from outside the A-tubule in **c** and along PF-A9 in **d**. MIP2 has 48-nm periodicity as previously shown[7,11] and mainly binds on PF-A9. MIP2 is shown to be composed of three different structures MIP2a, MIP2b and MIP2c instead of two types as previously proposed[7,11]. **(e)** View showing lateral interactions among MIP2, MIP4 and fMIP-A11A12 densities. MIP4 shows 48-nm periodicity as previous studies[7,11] and has four different types with a variety of shapes (called MIP4a, b, c, and d here). **(f)** View showing interactions between MIP2a/4c and MIP7 through the A-tubule lattice. One repeating unit of MIP7 is indicated by a bracket. **(g)** Density map showing interactions between MIP2/4 and MIP7 through the A-tubule lattice. Two of four MIP7 in 48-nm unit are connected to MIP2/4. Section is indicated by black line in the diagram. The structures shown in Fig. 6 are from 48-nm averaged density map. Red arrowheads in **a–d** indicate the part of MIPs protruding into the tubulin lattice. White arrowheads in **a** and **e** show lateral interactions between MIPs. Blue arrowheads in **a,b,f** and **g** indicate the interactions between MIP2/4 and MIP7 through the A-tubule lattice. ( + ) and ( − ) in **c–g** show the polarity of PFs and brackets in **c** and **e** indicate each tubulin dimer.

MIPs induces distortion of the A-tubule. MIP7 binds at the proper location possibly using insertion of MIP2/4 as landmarks (Fig. 8b, (ii)). At the same time, the tubulin molecules bind outside of the microtubule lattice by non-canonical tubulin–tubulin interaction shown here. The local distortion in the A-tubule is thought to facilitate the specific formation of the outer junction. MIP7 might also facilitate the recruitment of tubulin between PFs-A10 and A11. As tubulin forms the B-tubule lattice by canonical tubulin–tubulin lateral contact (Fig. 8b, (iii)), fMIPs bind to this region to ensure the B-tubule forms a rigid structure so PF-B10 can interact with MIP3 and the B-tubule can close properly at the inner junction side (Fig. 8b, (iv)). Our proposed model is consistent with the previous observation showing the B-tubule-like hook can elongate by itself, but rarely closes without the help of other proteins[20].

Having associated proteins with different periodicities such as MIPs (16- and 48-nm), outer dynein arms (24-nm) and radial spokes (96-nm), bindings of these proteins on the doublets are coordinated. Thus, there must be cooperativity in the assembly of MIPs and outside proteins, probably mediated by extensive lateral contacts between MIPs. Such interactions between the outside proteins and MIPs should be observed in the intact doublet in the future study.

Our high-resolution structure of the doublet provides many new insights into the unique assembly of the doublet even though the information about the identities and localizations of MIPs is limited at this point. To address the identities of the MIPs, structural studies of protein tagging and knockout mutants should be carried out or high-resolution structures of MIPs obtained by other structural techniques that can be docked into

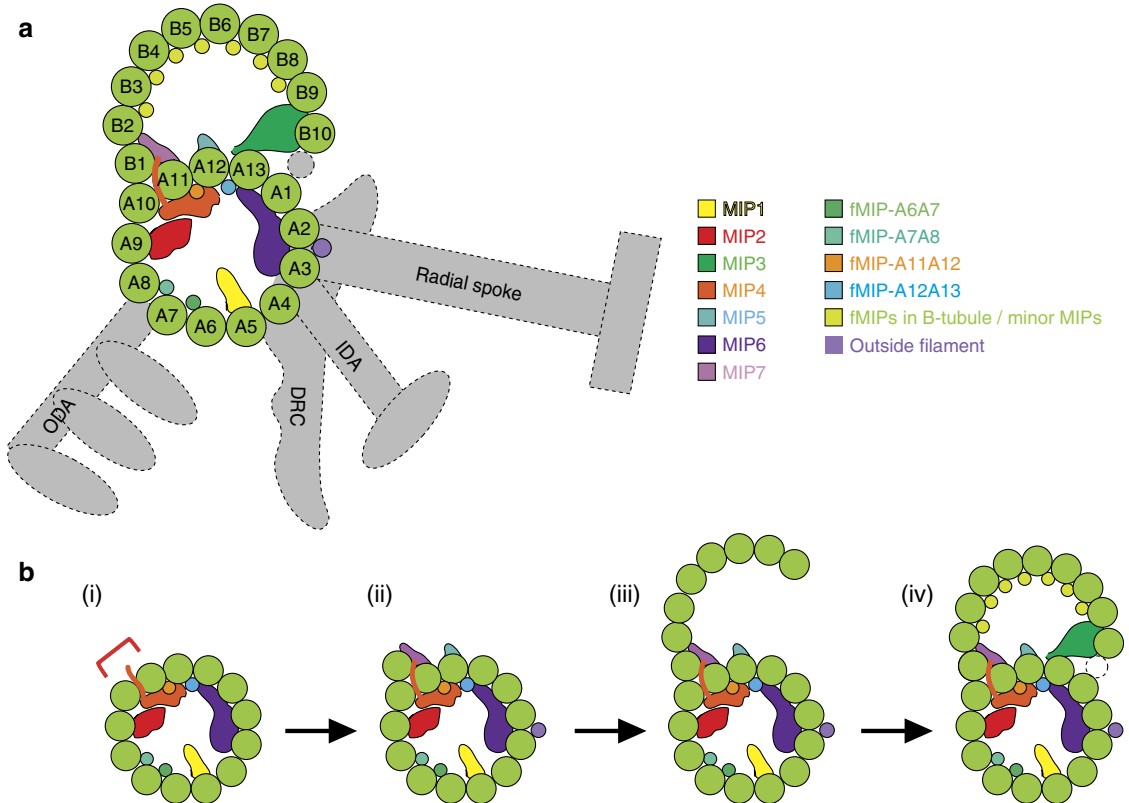

**Figure 8 | Possible assembly process of the doublet. (a)** Cartoon depicting the location of globular MIPs, fMIPs and outside proteins binding to *Tetrahymena* doublet. The structures with the dotted line are not observed in our structure. (**b**) A proposed assembly sequence of the doublet. (i) MIPs bind inside the A-tubule during its construction, causing the distortion in the A-tubule. PF pairs with red bracket indicates the PF-A10 and A11 which has a favourable angle for the outer junction formation and MIP2/4 insertion. (ii) Non-canonical binding of tubulin of PF-B1 to PFs-A10 and A11 is formed and further stabilized by MIP7. The B-tubule lattice starts to form (iii) and stabilized by fMIPs in the B-tubule so that it can be closed by interacting with MIP3 and possibly with the non-tubulin inner junction structure (iv).

our doublet structure. It is also interesting to compare our structure to a high-resolution structure of the doublet from motile and non-motile cilia, and from other species to reveal the conserved and essential MIPs. Finally, there is still a need for the structure of the doublet at near-atomic resolution to address the exact interaction interface and the interactions between the inner and outer MAPs.

## Methods

**Purification of doublet.** *T. thermophila* SB255 strain (mucocyst-deficient strain) was purchased from *Tetrahymena* Stock Center (Cornell University) and cultured in SPP media (1% proteose peptone No.3, 0.2% glucose, 0.1% yeast extract, 0.003% Fe-EDTA). Deciliation was performed by the dibucaine method[32]. Cell bodies were removed by low-speed centrifuge (2,000*g*, 7 min) and cilia were collected by high-speed centrifugation (17,000*g*, 30 min). Collected cilia were resuspended in the cilia final buffer (CFB; 50 mM Hepes, pH 7.4, 3 mM MgSO$_4$, 0.1 mM EGTA, 0.5% Trehalose, 1 mM DTT), demembraned with 1.5% NP-40, and doublets were split by adding 0.4 mM ATP (ref. 15). To obtain core doublets, split doublets were incubated with CFB containing 600 mM NaCl for 30 min on ice twice to deplete dynein arms and then dialyzed against HDM buffer (5 mM HEPES, pH 7.4, 1 mM DTT, 0.5 mM EDTA) to eliminate radial spokes[33].

**Electron microscopy.** The doublet is known to have preferred orientation in the cryo-EM (ref. 15). To get more random orientation of doublets on the cryo-EM grids, core doublets were sonicated and resuspended in CFB containing 600 mM NaCl (Supplementary Fig. 1b). 3.5 µl of sonicated core doublets were applied to glow-discharged holey carbon grids (Quantifoil R2/2), blotted and frozen in liquid ethane using the Vitrobot Mark IV (FEI Company).

Movies of the doublets were obtained at 59kx nominal magnification on the direct electron detector Falcon II with the FEI Titan Krios using a total dose of 45 electrons/Å$^2$ and 7 frames (calculated pixel size of 1.395 Å/pixel). The defocus range was between − 1.2 and − 3.8 µm. Pixel size was corrected to 1.375 Å/pixel by fitting the tubulin model[16] to our 4.6 Å PF map (see later section).

**Image processing.** The movies were motion corrected[34] and the contrast transfer function of the sum micrographs were estimated using CTFFIND4 (ref. 35). After discarding micrographs with apparent bad contrast transfer function estimation, drift and ice contamination, 5,983 micrographs were selected. Doublets in the micrographs were picked by e2helixboxer[36], which yielded 127,429 8-nm particles, 64,116 16-nm segments and 15,697 48-nm segments.

The data were initially processed using a modified version of the Iterative Helical Real Space Reconstruction script[37] in SPIDER[38] to work with non-helical symmetry. The data was then converted and processed with RELION[39]. Doublets were obtained with 8-, 16- and 48-nm repeat at resolutions of 5.7, 6.2 and 8.6 Å, respectively. Despite having all the angular orientations, preferred orientation might still limit our resolutions despite having a large number of particles.

After identification of α- and β-tubulins in the A- and B-tubules (see later section), subvolumes centring in the middle of tubulin dimers from each PF were boxed out from the density map of 8-nm repeat. The boxes containing tubulin PFs (13 from the A-tubule and 10 from the B-tubule) were aligned and averaged using subtomogram averaging procedure[40] without imposing a missing wedge (Supplementary Fig. 1f). The resulting PF map reached a resolution of 4.6 Å. The map was sharpened using Relion with a B-factor of − 180.

**Building of *Tetrahymena* tubulin model.** For building an initial model of the tubulin dimer, the *T. thermophila* sequences were extracted from UNIPROT and models were generated with MODELLER[41], using the *Sus scrofa* tubulin (PDB: 3JAR)[16] as a template. Subsequent flexible optimization of the interface was performed with HADDOCK 2.2 (ref. 42). A total of 20 structures were generated for each complex. Non-bonded interactions were calculated with the OPLS force field[43] using a cutoff of 8.5 Å. The electrostatic potential (Eelec) was calculated by using a shift function while a switching function (between 6.5 and 8.5 Å) was used to define the Van der Waals potential (Evdw). Desolvation energy was calculated as previously described[44]. Clustering at 7.5 Å was performed with the Rodrigues et al., contact-based function[45]. The tubulin dimer model was further refined by PHENIX.refine[17] using the PF density map of 4.6 Å resolution. The refinement strategy employed in our study was the minimization_global with five macro cycles and the default refinement parameters. As we found that the tubulin dimers in PF-B1 in our 8 nm-repeat density map have different conformations at the

M-loop region, we further refined tubulin model above by PHENIX in the PF-B1 region of the 8-nm repeat density map.

**Distinguishing α- and β-tubulins in our density map.** To evaluate the assignment of α- and β-tubulins, we first distinguished α- and β-tubulins in our 8-nm density map by visual inspection. The visual inspection was done based on the superimposition of initial model structures of *Tetrahymena* α- and β-tubulins; we found that the region (aa 36–50) is the most distinguishable part. This region not only shows two different secondary structures as a loop and an α-helix for α- and β-tubulins, it also presents the obvious difference in the density map (Supplementary Fig. 1e). Two persons did the inspection independently and the assignments were identical. Fitting was also evaluated by cross-correlation analysis by shifting tubulin dimers by 4-nm for each PF.

**Building of PF model and rotation angle calculation.** A model of a PF was generated by fitting *Tetrahymena* tubulin model structure into densities of consecutive tubulin dimers in the 4.6 Å PF density map. The PF model was then rotated and fitted into each PF in the doublet density map.

Since the local curvature of the doublet at different PFs is related to the rotation angle between tubulin dimers of adjacent PFs (Supplementary Fig. 2a), we used the UCSF Chimera[46] 'measure' tool to calculate the rotation angle and Z-shift between every pair of adjacent PF models. To avoid spurious results from fitting, we made sure the rotation axis is parallel to the microtubule longitudinal axis in every calculation. Theoretical microtubule PF numbers were calculated by dividing 360° with rotation angles. The results of the calculation are shown in Supplementary Table 1 and Supplementary Fig. 2b.

**Interaction surface analysis.** To analyse the interaction surface at the outer junction, we analysed six lateral interfaces between A10, A11 and B1 by energy minimization with HADDOCK. The six interfaces are (1) A10 α—A11 α; (2) A10 β—A11 β (canonical tubulin interaction), (3) A10 α—B1 α; (4) A10 β—B1 β; (5) A11 α–B11 α; (6) A11 β—B1 β. We also compared these six interfaces to the longitudinal interface of α- and β-tubulins within and between dimers (intra and inter dimer interactions). The results of the analysis are in Supplementary Table 2. Fitted tubulin model structure was analysed using PDBePISA (ref. 19) and tubulin residues forming salt bridges were estimated.

Hypothetical outer junction structures on the other PF pairs in the A-tubule were generated as follows. Since tubulins from A11 and B1 are connected by MIP7, we assume that the relative position between PFs-A11 and B1 should be fixed. Therefore, the outer junction model structure was divided into two regions: PF-A10 and rigid complex of PFs-A11/B1. To generate each hypothetical structure between a PF pair, A10-PF was fitted to one PF and PFs-A11/B1 complex is fitted to the adjacent PF by rigid body fitting using only PF-A11 region (Supplementary Fig. 2f). The generated hypothetical models were evaluated by the salt bridge formation between A10-tubulin and B1-tubulin using PDBePISA (ref. 19).

**Segmentation.** To identify all the densities of doublet-associated proteins, we filtered our doublet tubulin lattice model to corresponding resolutions and subtracted those densities from the corresponding EM density maps to create difference maps (16- and 48-nm maps; Supplementary Fig. 3a). The subtracted densities were then coloured manually using Chimera[46] and segmented based on colours. In the difference maps, we can visualize all the MIPs previously described in the literatures at sub-nanometre resolution and identify additional MAP densities (Supplementary Table 3).

**Mass spectrometry.** For mass spectrometry, we performed in-gel digestion. Gel bands were processed using standard methods[47]. The resulting peptides were loaded onto a Thermo Acclaim Pepmap (Thermo, 75 μm ID × 2 cm C18 3 μm beads) precolumn and then onto an Acclaim Pepmap Easyspray (Thermo, 75 μm × 25 cm with 2 μm C18 beads) analytical column separation using a Dionex Ultimate 3000 μHPLC at 200 nl min$^{-1}$ with a gradient of 2–35% organic (0.1% formic acid in acetonitrile) over 2 h. Peptides were analysed using a Thermo Orbitrap Fusion mass spectrometer operating at 120,000 resolution (FWHM in MS1, 15,000 for MS/MS) with HCD sequencing all peptides with a charge of 2+ or greater. The raw data was converted into *.mgf format (Mascot generic format) for searching using the X!Tandem search engine (Beavis Informatics) against human open reading frames (Uniprot). The database search results were loaded onto Scaffold Q+ Scaffold_4.4.8 (Proteome Sciences) for statistical treatment and data visualization.

Among proteins identified, we detected homologues of Rib72 (UniprotId I7M0S7 & I7MCU1), Rib43a (A4VDZ5 & Q240R7), FAP20 (Q22NU3), PACRG (I7M317 & I7MLV6), FAP59 (Q23BW0) and FAP172 (Q233L0).

**Data availability.** EM reconstructions and refined tubulin models are available in the Electron Microscopy Data Bank (EMDB) and Protein Data Bank (PDB) with following accession numbers: PF (EMD-8539, PDB: 5UCY), 8-nm (EMD-8528, PDB: 5UBQ), 16-nm (EMD-8532) and 48-nm repeat maps (EMD-8537). The

datasets analysed during the current study are available from the corresponding author upon reasonable request.

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

## Acknowledgements

We are indebted to Drs Masahide Kikkawa, Takashi Ishikawa, Haruaki Yanagisawa, Susanne Bechstedt, Gary Brouhard, S. Kelly Sears and Martin Beck for their critical reading of the manuscript and insightful discussions. We are grateful to Drs Justin Kollman, Kalle Gehring, Hojatollah Vali, Craig Mandato and Eric Boucher for indispensable advises and research support. We also acknowledge the support of the Facility for Electron Microscopy Research and the McGill University Health Centre Proteomics platform, in particular, Dr S. Kelly Sears and Ms. Amy Ho Yee Wong. We thank NVIDIA for their generous Academic GPU grant. M.I. is supported by the Dr David T.W. Lin Fellowship from McGill University. This research was supported by funding from Natural Sciences and Engineering Research Council (NSERC) of Canada (Discovery Grant 69462) and startup funds from McGill University to K.H.B.

## Author contributions

K.H.B. and M.I. designed the study and M.I. performed cell culture, purification, preparation of cryo-EM grids and mass spectrometry preparation with helps from T.C.H. and S.Y.; M.I. and K.B. performed cryo-EM data acquisition. M.I., D.L. and K.H.B. performed EM data processing, structure refinement and interpretation. P.L.K performed initial structural modelling and energy calculation. K.H.B, M.I. and D.L. wrote the manuscript.

## Additional information

**Competing interests:** The authors declare no competing financial interests.

