## [Peer Review File · Nature Communications]

Reviewers' Comments:

Reviewer #1 (Remarks to the Author)

This manuscript reports a structural study of microtubule doublets from motile cilia of *Tetrahymena* at sub-nanometer resolutions. The reconstruction was achieved with single particle approach using the IHRSR algorithm, followed by a Relion refinement. The resolution of the density map is the highest among the published maps by far, with secondary structural features clearly visible. The high quality of the map has enabled the author to build a plausible structural model by rigid body fitting. Based on the structural model, a generated difference map revealed some filamentous density domains, along or in between the microtubule protofilaments (PFs). The manuscript reports them as a new class of proteins among the microtubule associated proteins (MAPs) that bind intralumenally to the microtubules (filamentous microtubule inner proteins: fMIPs). In addition to the filamentous densities, there are several other pieces of new information in the manuscript that are important in advancing the understanding of microtubule doublet structure: (1) the location of the seam in the complete A tubule is between the PF A9 and the PF A10; (2) The outer junction and the inner junction structures between A and B tubule are better resolved. Overall, this manuscript contains important new structural information and should be published after the authors make a better presentation of the data to convince general readers to appreciate the new results.

There is some room for the authors to improve their result presentation, and some of the issues are on technical aspects.

(1) The resolution:

I am not able to make sense out of the FSC profile in Fig S-1C. First of all, it did not show the units of the axes. Since the range of the x axis is from 0 to 0.5, we could assume the x-axis is of a popular frequency unit \AA^{-1} . If so, the 0.143 FSC would indicate a resolution of $\sim 2\text{\AA}$ for the 8nm repeats (red) and 16nm repeats (green), and $\sim 3\text{\AA}$ for the 48nm repeats. This cannot be true for the map quality shown in the Figure 1 (I will talk about it later). Even after I take the pixel size of 1.395\AA into account, the FSC profiles is not consistent with the resolution values in the text. Sometimes, artifacts do happen in FSC calculation and lead to a wrong resolution estimation. However, two of the three FSC curves in Fig S-1C were cut off before they reach zero, and makes it difficult for me to evaluate the reliability of this FSC calculation.

To estimate the resolution and the structural map quality, I carefully looked into the enlarged

figures for Figure 1A, 1C and 1D. In the magnified density maps in Figure 1, some sausage-like features are visible that have been fitted to the helices of tubulins, although not all of them fit well. Such level of structural details indicates a resolution beyond 10~12Å. However, probably due to the way the authors made the Figure, I failed to see the pitch features of alpha helices, which should easily be visible for maps with a resolution beyond 6Å. Considering the above, I suggest the authors re-evaluate the resolutions they have claimed.

I would like to point out that, while some readers in structural biology take the namely value of resolution a big deal, most mature structural biologists do not. Most senior or experienced structural biologists would pay more attention to the structural features disclosed by the density map, such as secondary structural features, helical pitches or big side chains of amino-acid residues, instead of the FSC value that is occasionally misleading. Whenever a claimed resolution number is given, the structural feature has to match the claimed resolution or buzz may rise among the readers. Therefore, the namely value of the highest resolution should not be a major factor for publication consideration.

(2) Model fitting for finding the seam:

(a) In the Figure 1b, it looks like that the fitting can be just as good, when/if the alpha and beta tubulin structural models are switched. Also, in this picture, it looks like the helix H11 of tubulin is totally out of the density maps for both alpha and beta tubulins. A better figure or a better fitting is needed to convince general readers that the current model is a good and unambiguous fitting.

(b) It is an important conclusion that the seam locates between PF9 and PF10 in A tubule. To make it better for readers to appreciate the result, the authors should replace the magnified A8 PF panel with a magnified seam region showing both PF9 and PF10, which will also demonstrate the credibility of the conclusion.

(3) About the curvature measurement:

Measuring the relative rotation angles between the neighboring PFs does provide useful information. However, to use the angles for measuring the “curvature” without a concept clarification can introduce confusion, because this is not entirely consistent with the traditional curvature understanding.

To make this point clear, please refer to the attached picture. The traditional curvature concept was introduced when the detailed FP structures were unavailable. For general readers, as shown in the attached Fig 1., the “curvature” at the point P of a plane smooth curve C, is defined as the inverse of the radius of the curvature of C at the point P, which is a measure of how sensitive its

tangent line is to moving the point to other nearby points. The curvature concept is useful for smooth curves instead of for abrupt turns. Traditionally, in a typical microtubule as shown in the attached Fig 2, the curve C (a round circle in this case), is generated by the molecular centers of tubulins. A rotation of the PF (7 degrees for the green block/PF in Fig 2 for example) does not necessarily change the curvature of curve C at the point P. Rather, without a further modification or refinement of the curvature concept, this rotation may be better described as a local PF rotation or structure conformational change instead of a PF-13 curvature change. You may imagine to replace this green PF with a different type filament (say tektin) without a curvature change, then a relative angle rotation can not be used to describe such curvature. Therefore, using the angles to determine the traditional curvature concept can lead to a confusion if without a re-definition of the wall curvature concept of microtubules.

(Fig 1 and Fig. 2 can not be inserted here. Please find them in the attachment)

Therefore, I am not sure it is a good idea to make the rotation angles equivalent to the classic curvature concept. This is particularly true when the Supplementary Fig 1d and Table S1 gives abrupt differences without predictable trend among some of the neighboring rotation angles. Such abrupt changes do not represent a smooth curve, and makes it questionable using the angles for “curvature” measurement.

On the other hand, there is nothing wrong if the authors insist to use the curvature path for structural information delivery. However, doing so, a clarification of the new/refined curvature concept is needed in the manuscript, otherwise some readers, such as myself, may become confused.

(4) There is some structural information in the supplemental text that are important enough to be included in the main text. In particular, the authors should consider to move the content about the structural architecture of the inner junction into the main text, and this will make the whole structure description of the microtubule doublet more complete. In fact, this information is of an important value, as it confirms that an extra non-tubulin filament at the inner junction region is not essential for structural integration of the microtubule doublet.

(5) Nowadays, resultant maps of published papers should be deposited into the EM Data bank. The authors probably have done so. Then, the deposition information should be included in the manuscript.

(6) Some other minor errors:

(a) Page 3, Paragraph 2, line 2: “...and fitted it into our EM density map (Fig. 1b).” should be “... and fitted into our EM density map (Fig. 1b).”, i. e., remove the “it”.

(b)Page 5, Paragraph 2, Line 2 “...four types lateral contacts...” should be “...four types of lateral

contacts...”, i.e., add a “of”.

(c) Page 8, Paragraph1, Line 7: “...and the B-tubule can close properly at inner junction...” should be “...and the B-tubule can close properly at the inner junction...”, i.e., add a “the”.

Reviewer #2 (Remarks to the Author)

The authors used cryo-electron microscopy to visualize the structure of the doublet microtubule, which is an essential component of the inner core structure of cilia called axoneme. The resolution of the 3D structure is high enough to show for the first time many different microtubule inner proteins (MIPs) bound to the inner surface of the doublet, which appear to stabilize the doublet microtubule structure.

Although the structure is impressively showing much more details than the previous studies, it is difficult to judge if the authors' claim of resolution, 5.7 Å to 8.6 Å, is reasonable and experimentally supported. The Fourier shell correlation curves in Supplementary Fig. 1c is incomplete, showing the curves only to the point where the FSC decays to 0.143, and the unit of Fourier space in x-axis is not designated, making it impossible to judge the resolution. The 3D density map at 5.7 Å resolution is supposed to allow identification of individual secondary structures, such as α -helices, β -sheets and loops, but the maps presented in Fig. 1 with models of α - and β -tubulins are all difficult to see whether individual secondary structures can be really identified, even in magnified ones shown in Fig. 1d and e. This also makes it hard to believe that the map allowed the authors to distinguish α -tubulins from β -tubulins in the structure.

The resolutions of density regions identified as MIPs look much worse than those of tubulins. No explanations are given for this, but that is why the map did not allow identification of any of the MIPs, limiting their structural descriptions just to their sizes, shapes and locations.

Overall, although this study certainly advances our understanding of how doublet microtubule structure is stabilized by all those MIPs binding, it is this reviewer's judgement that the biological implications and insights obtained from this structure are rather limited.

Minor points for improvement.

1. Abstract, line 7: It is not “in situ” structure.
2. p. 8, line .8: There is no “iv” in Fig. 5b.
3. Fig. 2: Displayed regions should be more accurately indicated in left panels in b to g.
4. Fig. 5: It would be easier for readers to follow the text if PFs are numbered.

Reviewer #3 (Remarks to the Author)

Ichikawa et al. High-resolution structure of the doublet microtubule ...

This is a beautiful piece of work, and the authors present a huge amount of data within this rather cramped format of Nature Communications. I cannot see any flaws within the data as produced by cryo-EM as well as the docking attempts of known near-atomic data, and most of the interpretations. The authors present a large variety of novel microtubule-inner binding proteins (MIPs) at a spectacular and very impressive resolution. MIPs have been shown before in microtubule doublets and other microtubule structures, but never at the resolution achieved here. In my opinion this work justifies publication in a journal with a reputation like Nature Comm.

My only criticism is about the overall presentation, and the amount of highly relevant data that went into the supplementary material section, despite numerous citations of these figures and tables in the main text. Often the supplementary data is not accessible everywhere, and missing it out would make the paper hard to fully appreciate, and somewhat incomplete. All of the figures show gorgeous data, but they are heavily overloaded and quite cramped, apparently a result of trying to get as much data as possible within the journal's limit on figures in the main text ... Therefore, because of these formatting issues I wonder slightly if Nature Communications is the best suitable venue for this paper. However, I do not necessarily see this as a no-go for publication, and I would leave it to the Editors to decide whether they accept this format or not.

A few points:

Cryo-EM does not allow distinguishing close proximity from actual binding. Therefore the theories about non-canonical A-tube with B and B with C contacts are a bit over-interpreted. Many dynein and kinesin motors and MAPs bind to all forms of tubulin, such as tight or wide tubes and flat tubulin sheets (also called c-tubules). The only tubulin form that does interact well with motors are zinc-induced sheets which show a different protofilament arrangement. The Zn-sheets also show a very different lateral protofilament interaction.

The actual shape of some MIPs may be distorted by the averaging procedure, required to reach such high resolution (i.e. some of the continuous filamentous MIPs may in fact be short stretches of protein mass, even if you take 48 nm steps). There is no way around averaging yet for such data. Hence, this should be mentioned somewhere to make a general audience aware of this, but this is no criticism on the data as such.

Response to reviewers' comments

Manuscript (NCOMMS-16-25589)

Title: High-resolution structure of the doublet microtubule reveals new classes of microtubule-associated proteins and insights into its assembly mechanism

We would like to thank all the reviewers for their suggestions and constructive criticisms. Upon addressing the reviewers' comments, we performed a major revision of our data analysis. We think the results from the new data analysis add to the significance of the manuscript and appropriately address the reviewers' concerns regarding model fitting and interpretation.

The new data analysis includes:

- Obtaining the density map of the protofilament at 4.2 Å resolution.
- Obtaining an accurate tubulin model by real space refining in the above map.
- Building a protofilament model from the high-resolution protofilament map for more accurate modelling in the doublet map
- Analysing the difference of tubulin conformation, in particular the M-loop in protofilament B1.

Reviewer #1 (Remarks to the Author):

This manuscript reports a structural study of microtubule doublets from motile cilia of Tetrahymena at sub-nanometer resolutions. The reconstruction was achieved with single particle approach using the IHRSR algorithm, followed by a Relion refinement. The resolution of the density map is the highest among the published maps by far, with secondary structural features clearly visible. The high quality of the map has enabled the author to build a plausible structural model by rigid body fitting. Based on the structural model, a generated difference map revealed some filamentous density domains, along or in between the microtubule protofilaments (PFs). The manuscript reports them as a new class of proteins among the microtubule associated proteins (MAPs) that bind intralumenally to

the microtubules (filamentous microtubule inner proteins: fMIPs). In addition to the filamentous densities, there are several other pieces of new information in the manuscript that are important in advancing the understanding of microtubule doublet structure: (1) the location of the seam in the complete A tubule is between the PF A9 and the PF A10; (2) The outer junction and the inner junction structures between A and B tubule are better resolved. Overall, this manuscript contains important new structural information and should be published after the authors make a better presentation of the data to convince general readers to appreciate the new results.

There is some room for the authors to improve their result presentation, and some of the issues are on technical aspects.

(1) The resolution:

I am not able to make sense out of the FSC profile in Fig S-1C. First of all, it did not show the units of the axes. Since the range of the x axis is from 0 to 0.5, we could assume the x-axis is of a popular frequency unit \AA^{-1} . If so, the 0.143 FSC would indicate a resolution of $\sim 2\text{\AA}$ for the 8nm repeats (red) and 16nm repeats (green), and $\sim 3\text{\AA}$ for the 48nm repeats. This cannot be true for the map quality shown in the Figure 1 (I will talk about it later). Even after I take the pixel size of 1.395 \AA into account, the FSC profiles is not consistent with the resolution values in the text. Sometimes, artifacts do happen in FSC calculation and lead to a wrong resolution estimation. However, two of the three FSC curves in Fig S-1C were cut off before they reach zero, and makes it difficult for me to evaluate the reliability of this FSC calculation.

The previous FSC profile was calculated based on Bin 2 data (pixel size 2.79 \AA). Therefore, the obtained resolutions of the 8nm- and 16-nm repeat map are close to Nyquist frequency (5.58 \AA). We reanalyzed using Bin 1 data (pixel size 1.395 \AA) and made the new FSC profile. The new FSC profile also includes our new protofilament map at 4.2 \AA resolution. We put the correct unit in the X-axis.

To calculate the FSC, we used soft edge masks and using Relion's FSC calculation with correction procedure to avoid effect of the edge. One difference is the resolution of the 16-nm map is now 6.2 \AA instead of 5.8 \AA before. The reason is that we make the soft edge mask softer and bigger to avoid inflation in FSC measurement.

To estimate the resolution and the structural map quality, I carefully looked into the enlarged figures for Figure 1A, 1C and 1D. In the magnified density maps in Figure 1, some sausage-like features are visible that have been fitted to the helices of tubulins, although not all of them fit well. Such level of structural details indicates a resolution beyond 10~12 \AA . However, probably due to the way the authors made the Figure, I failed to see the pitch features of alpha helices, which should easily be visible for maps with a

resolution beyond 6Å. Considering the above, I suggest the authors re-evaluate the resolutions they have claimed.

Although the overall resolution of the 8-nm map is 5.7 Å, the resolution is not isotropic. We included now more details of the local resolution estimation by ResMap to clear the confusion. For example, the region of protofilaments A10-A13 has the highest resolution while other regions, in particular the B-tubule, has lower resolution (up to 9-10 Å).

We updated the magnified density map in Fig. 1b to reflect the structural features expected to be seen at the reported resolution by reducing the threshold level in surface rendering. Before, we tried to use a threshold to display the entire map well, but because of the anisotropy of resolution, it makes regions like PF-A10 look bad.

We also added this sentence in the text to talk about the anisotropy of our resolution:

“Local resolution analysis shows that, the A-tubule was resolved better compared to the B-tubule, especially at the ribbon region (the junction site of the A- and B-tubules) (**Supplementary Fig. 1d**), reflecting the high stability of this region.”

I would like to point out that, while some readers in structural biology take the namely value of resolution a big deal, most mature structural biologists do not. Most senior or experienced structural biologists would pay more attention to the structural features disclosed by the density map, such as secondary structural features, helical pitches or big side chains of amino-acid residues, instead of the FSC value that is occasionally misleading. Whenever a claimed resolution number is given, the structural feature has to match the claimed resolution or buzz may rise among the readers. Therefore, the namely value of the highest resolution should not be a major factor for publication consideration.

As mentioned above, we included now an update Fig. 1c to show clear α - helices in our 8-nm map. To illustrate the resolution of our protofilament map, we now include images showing density of bulky residue in Fig. 1b.

(2) Model fitting for finding the seam:

(a) In the Figure 1b, it looks like that the fitting can be just as good, when/if the alpha and beta tubulin structural models are switched. Also, in this picture, it looks like the helix H11 of tubulin is totally out of the density maps for both alpha and beta tubulins. A better figure or a better fitting is needed to convince general readers that the current model is a good and unambiguous fitting.

We now clarified the procedure to distinguish α - and β -tubulins in our method and Supplementary Fig.1e. We used both cross correlation as in Maheshwari *et al.*, 2015 and

also visual inspection of regions of aa36-50 of α - and β -tubulins. Our results from both methods agreed with the result from Maheshwari *et al.*, 2015 in which α - and β -tubulin lattices of some protofilaments have been confirmed by kinesin decoration, except PFs-A1, A3, A6, A7, A10-A13. Since the resolutions in these protofilaments in our density map are good, we can unmistakably determine α - and β -tubulins by our procedures.

One thing to note is that the residues predicted to be involved in salt bridge formation has changed slightly in our new doublet model. We believed that it is a result of more accurate modeling and fitting in our analysis. In the new analysis, we improved tubulin model (refined in 4.2 Å PF map), pixel calibration (1.375 Å instead of 1.395 Å), PF model (fitted in 4.2 Å PF map) and tubulin model to the specific conformation from PF B1.

(b) It is an important conclusion that the seam locates between PF9 and PF10 in A tubule. To make it better for readers to appreciate the result, the authors should replace the magnified A8 PF panel with a magnified seam region showing both PF9 and PF10, which will also demonstrate the credibility of the conclusion.

We put the magnified picture of the seam in Fig. 1c.

(3) About the curvature measurement:

Measuring the relative rotation angles between the neighboring PFs does provide useful information. However, to use the angles for measuring the “curvature” without a concept clarification can introduce confusion, because this is not entirely consistent with the traditional curvature understanding.

To make this point clear, please refer to the attached picture. The traditional curvature concept was introduced when the detailed FP structures were unavailable. For general readers, as shown in the attached Fig 1., the “curvature” at the point P of a plane smooth curve C, is defined as the inverse of the radius of the curvature of C at the point P, which is a measure of how sensitive its tangent line is to moving the point to other nearby points. The curvature concept is useful for smooth curves instead of for abrupt turns. Traditionally, in a typical microtubule as shown in the attached Fig 2, the curve C (a round circle in this case), is generated by the molecular centers of tubulins. A rotation of the PF (7 degrees for the green block/PF in Fig 2 for example) does not necessarily change the curvature of curve C at the point P. Rather, without a further modification or refinement of the curvature concept, this rotation may be better described as a local PF rotation or structure conformational change instead of a PF-13 curvature change. You may imagine to replace this green PF with a different type filament (say tektin) without a curvature change, then a relative angle rotation can not be used to describe such curvature. Therefore, using the angles to determine the traditional curvature concept can lead to a confusion if without a

re-definition of the wall curvature concept of microtubules.

(Fig 1 and Fig. 2 can not be inserted here. Please find them in the attachment).

Therefore, I am not sure it is a good idea to make the rotation angles equivalent to the classic curvature concept. This is particularly true when the Supplementary Fig 1d and Table S1 gives abrupt differences without predictable trend among some of the neighboring rotation angles. Such abrupt changes do not represent a smooth curve, and makes it questionable using the angles for “curvature” measurement.

On the other hand, there is nothing wrong if the authors insist to use the curvature path for structural information delivery. However, doing so, a clarification of the new/refined curvature concept is needed in the manuscript, otherwise some readers, such as myself, may become confused.

This is an important point that reviewer 1 has pointed out. Since the tubulin interacts laterally with adjacent tubulin through the highly conserved M-loop and H1'-S2 and H2-S3 loops of adjacent tubulin, this interaction serves as a hinge between a pair of protofilaments. Therefore, the rotation angle between two adjacent protofilaments is directly related to the curvature. To define this concept, we make a panel in Supplementary Fig. 2a to illustrate our point of using the rotation angle to measure microtubule curvature. In addition, we also explained the concept in the text as follow:

“Local curvature of the microtubule lattice is in an inverse relationship with the rotation angle between neighboring tubulins since the regions of H1'-S2 and H2-S3 loops and the M-loop of the adjacent tubulin mediate the lateral interaction^{16,18} and act as the hinge between neighboring tubulins (**Supplementary Fig. 2a**). Therefore, we measured the rotation angles between fitted tubulin dimers in successive PFs and compared them to singlet microtubules with different PF numbers.”

We also put the following text in the caption of Supplementary Fig. 2a:

“Relationship between curvature of microtubule lattice and rotation angle of neighboring tubulin molecules. For high curvature microtubule (small number of PFs), absolute value of rotation angle of tubulin is larger with smaller radius (R). For low curvature microtubule (large number of PFs), absolute value of rotation angle of tubulin is smaller with larger radius (R').”

In the discussion, we also tried to use the word “local angle” instead of local curvature.

(4) There is some structural information in the supplemental text that are important enough to be included in the main text. In particular, the authors should consider to move

the content about the structural architecture of the inner junction into the main text, and this will make the whole structure description of the microtubule doublet more complete. In fact, this information is of an important value, as it confirms that an extra non-tubulin filament at the inner junction region is not essential for structural integration of the microtubule doublet.

We moved the part of the inner junction and some MIPs from supplementary text to the main text.

(5) Nowadays, resultant maps of published papers should be deposited into the EM Data bank. The authors probably have done so. Then, the deposition information should be included in the manuscript.

We deposited our maps of protofilament, 8nm, 16nm, and 48-nm to EMDB as well as the coordinate of tubulins in the protofilaments and 8nm to the PDB. The accession codes are now written in the paper.

(6) Some other minor errors:

(a) Page 3, Paragraph 2, line 2: "...and fitted it into our EM density map (Fig. 1b)." should be "... and fitted into our EM density map (Fig. 1b).", i. e., remove the "it".

(b)Page 5, Paragraph 2, Line 2 "...four types lateral contacts..." should be "...four types of lateral contacts...", i.e., add a "of".

(c) Page 8, Paragraph1, Line 7: "...and the B-tubule can close properly at inner junction..." should be "...and the B-tubule can close properly at the inner junction...", i.e., add a "the".

We fixed these points.

Reviewer #2 (Remarks to the Author):

The authors used cryo-electron microscopy to visualize the structure of the doublet microtubule, which is an essential component of the inner core structure of cilia called axoneme. The resolution of the 3D structure is high enough to show for the first time many different microtubule inner proteins (MIPs) bound to the inner surface of the doublet, which appear to stabilize the doublet microtubule structure.

Although the structure is impressively showing much more details than the previous

studies, it is difficult to judge if the authors' claim of resolution, 5.7 Å to 8.6 Å, is reasonable and experimentally supported. The Fourier shell correlation curves in Supplementary Fig. 1c is incomplete, showing the curves only to the point where the FSC decays to 0.143, and the unit of Fourier space in x-axis is not designated, making it impossible to judge the resolution. The 3D density map at 5.7 Å resolution is supposed to allow identification of individual secondary structures, such as α -helices, β -sheets and loops, but the maps presented in Fig. 1 with models of α - and β -tubulins are all difficult to see whether individual secondary structures can be really identified, even in magnified ones shown in Fig. 1d and e. This also makes it hard to believe that the map allowed the authors to distinguish α -tubulins from β -tubulins in the structure.

We addressed the same concern from reviewer 1 above. We included Supplementary Fig. 1e and text in Methods section to explain the procedure of distinguish α - and β -tubulins in the structure.

The resolutions of density regions identified as MIPs look much worse than those of tubulins. No explanations are given for this, but that is why the map did not allow identification of any of the MIPs, limiting their structural descriptions just to their sizes, shapes and locations.

It is true that the resolution of MIP regions are lower compared to the tubulin lattice. It is probably due to the flexibility of the MIPs and the doublet itself. To make it clear in the manuscript, we put in the local resolution estimation part, a section showing the local resolution estimation of some MIPs and corresponding tubulin regions (Supplementary Fig. 1d).

We also included the following text in the “Results” to address that issue in the text. “The local resolutions of the MIPs seem to vary from the overall resolution, probably due to flexibility (Supplementary Fig. 1d).”

Minor points for improvement.

1. Abstract, line 7: It is not “in situ” structure.

We removed the word “in situ”.

2. p. 8, line .8: There is no “iv” in Fig. 5b.

We fixed it and the numbering of figure and the text is consistent now.

3. Fig. 2: Displayed regions should be more accurately indicated in left panels in b to g.

We fixed the figure and explain it in the figure legend.

4. Fig. 5: It would be easier for readers to follow the text if PFs are numbered.

PFs in Fig. 5a (currently Fig. 7a) is numbered as suggested.

Reviewer #3 (Remarks to the Author):

Ichikawa et al. High-resolution structure of the doublet microtubule ...

This is a beautiful piece of work, and the authors present a huge amount of data within this rather cramped format of Nature Communications. I cannot see any flaws within the data as produced by cryo-EM as well as the docking attempts of known near-atomic data, and most of the interpretations. The authors present a large variety of novel microtubule-inner binding proteins (MIPs) at a spectacular and very impressive resolution. MIPs have been shown before in microtubule doublets and other microtubule structures, but never at the resolution achieved here. In my opinion this work justifies publication in a journal with a reputation like Nature Comm.

My only criticism is about the overall presentation, and the amount of highly relevant data that wet into the supplementary material section, despite numerous citations of these figures and tables in the main text. Often the supplementary data is not accessible everywhere, and missing it out would make the paper hard to fully appreciate, and somewhat incomplete. All of the figures show gorgeous data, but they are heavily overloaded and quite cramped, apparently a result of trying to get as much data as possible within the journal's limit on figures in the main text ... Therefore, because of these formatting issues I wonder slightly if Nature Communications is the best suitable venue for this paper. However, I do not necessarily see this as a no-go for publication, and I would leave it to the Editors to decide whether they accept this format or not.

We made a big effort to improve the presentations of the manuscript. We also split Fig. 1 into two figures now. And also move some panels from Supplementary Figure into the main figures about the interaction of globular MIPs. Therefore, we have now two figures related to the globular MIPs (Figs. 5 and 6).

We also moved the part of the inner junction and some MIPs from supplementary text to

the main text.

A few points:

Cryo-EM does not allow distinguishing close proximity from actual binding. Therefore the theories about non-canonical A-tube with B and B with C contacts are a bit over-interpreted. Many dynein and kinesin motors and MAPs bind to all forms of tubulin, such as tight or wide tubes and flat tubulin sheets (also called c-tubules). The only tubulin form that does interact well with motors are zinc-induced sheets which show a different protofilament arrangement. The Zn-sheets also show a very different lateral protofilament interaction.

It is certainly true that cryo-EM does not confirm any actual binding. Using a combination of modeling and biophysical tools, we tried to predict the non-canonical binding which might be a good target for future experiments.

We put the picture of the structure of the triplet from *Chlamydomonas* (Li *et al*, 2010, EMBO J) in Supplementary Fig. 2e to illustrate our point.

There are also proteins known to bind to specific curvature of microtubules and influence the assembly of 13-pf microtubule. We add this sentence to the text to reflect that:

"This is the reminiscence of microtubule associated proteins like EB1 and doublecortin which prefer to bind to the 13-PF microtubule^{20,21}."

The actual shape of some MIPs may be distorted by the averaging procedure, required to reach such high resolution (i.e. some of the continuous filamentous MIPs may in fact be short stretches of protein mass, even if you take 48 nm steps). There is no way around averaging yet for such data. Hence, this should be mentioned somewhere to make a general audience aware of this, but this is no criticism on the data as such.

We also put sentences to explain about the repeating unit of fMIPs "Since these features are visible only in 48-nm map, it is highly likely that these fMIPs are in 48-nm repeating unit except fMIP-A12A13 which appears similar in both 48-nm and 16-nm maps (**Supplementary Fig. 4b** and **Supplementary Table 4**)."

There is still a possibility that these fMIPs are in 96-nm repeating unit, so we also specify that possibility in the manuscript.

Reviewers' Comments:

Reviewer #1 (Remarks to the Author):

The authors provided newly updated data processed without binning which showed a significant improvement on data quality, and made it particularly convincing about the seam location. The authors have also provided updated Fourier shell correlation (FSCs). Most of the FSC profiles look reasonable except for the PF one, which shows significant raise-ups at the high frequency regions. It indicates this FSC profile would not be reliable for a resolution estimation. Therefore, the authors may want to be careful with the claim of the 4.2Å resolution based on this FSC. I would like to suggest to revisit the FP FSC calculation. Other than that, the revised manuscript represents a significant improvement, and I believe it can be published.

There are also some minor errors in the manuscript as following:

Page 4, paragraph 1, line 3: "... fitted well in doublet map..." should be "...fitted well in the doublet map...".

Page 4, paragraph 3, line 1: "...local curvature of doublet lattice..." should be "...local curvature of the doublet lattice...".

Page 5, paragraph 3, line 8: "...showing that doublet can be formed without the inner junction structure" should be "...showing that doublets can be formed without the inner junction structure ...".

Page 6, last paragraph, line 4: "...also show insertions into tubulin lattice..." should be "...also show insertions into the tubulin lattice ...".

Page 7, paragraph 6, 1st line: "MIP6 appears to have in fact 16-nm repeating unit..." should be "MIP6 appears to have in fact a 16-nm repeating unit...".

Page 7, last paragraph, 1st line: "We also observed 16-nm repeating unit density..." should be "We also observed a 16-nm repeating unit density...".

Page 8, 2nd paragraph, line 5: "...two tubulin dimers and between α -..." should be "...two tubulin dimers and between α -..." i.e. : needs a space between the "between α -"

Page 9, paragraph 3, line 4: "...with an earlier study showed that..." should be "...with an earlier study which showed that..."

Page 12, 1st paragraph, line 3: "...difference biophysical effects..." should be "...different biophysical effects..." : i.e., change "difference" to "different".

Page 18, paragraph 3, line 8: "The raw data were converted into..." should be "The raw data was converted into...".

Reviewer #2 (Remarks to the Author):

The revised manuscript is much improved in presentation of the structural details and the FSC for resolution. Only one remaining concern is that it is still difficult to see if α -tubulin and β -tubulin are well distinguished, especially at the seam between A9 and A10 shown in the right panel of Fig. 1c. What I would do to make it clearer is to present only the density map in the right panel of Fig. 1c without the fitted model and indicate a few density features with which readers can easily distinguish α -tubulin from β -tubulin. Also, Supplementary Fig. 1e should be magnified make it easy for readers to see the differences between α -tubulin and β -tubulin.

Reviewer #3 (Remarks to the Author):

Ichikawa et al.: High-resolution structure of the doublet microtubule:

This paper shows cryo-EM reconstructions of microtubule doublets and thereby identifies a large number of so-called microtubule inner proteins (MIPs) who have been seen before, but no such detailed analysis has been reported yet. The authors exploit the newest technologies on the hardware (e.g. direct electron detectors) and the software sides, and present a stunning piece of work.

I was already very positive about this work the first time I saw it. The paper certainly improved even more due to the reviewer's comments. I almost wonder if this work would not give sufficient material for two publications ... As the first version, this is still an extremely overloaded paper with huge amounts of supplementary data, but if the editors are fine with this format I do not see any reason to delay publication. This is an important piece of data for a very active field of research, and even if some claims (e.g. the overall resolution of 0.62 nm at the very challenging 3sigma cutoff) are a bit too optimistic, especially for the MIP structures, this is a solid work. Useful resolution improves in steps, and often a range from 6-8Å is not really showing anything new but only sharpen edges. E.g. to really see the pitch of α -helices (as one reviewer pointed out) better resolution will be necessary. High resolution claims of MIP structures are not adding much to the data and may be corrupted by averaging artefacts, but the

important part is that the authors were able to resolve and identify all the different MIPs presented. The addition of density images greatly helped to make the story more believable, at least for old-school microscopists as myself. I recommend this work for publication, provided the data has been properly uploaded to a 3-D EM database, and the editors have no problem with the cramped data presentation.

Response to reviewers' comments

Manuscript (NCOMMS-16-25589A)

Title: Subnanometer-resolution structure of the doublet microtubule reveals new classes of microtubule-associated proteins

We would like to thank all the reviewers for their suggestions and constructive criticisms again. We tried to address all the suggestions from the reviewers and also the editor in this final revision. All other changes are highlighted in the manuscripts with the Track Change feature of Microsoft Word.

The notable changes are:

- We changed the title into “*Subnanometer-resolution structure of the doublet microtubule reveals new classes of microtubule-associated proteins*” to conform with 15-word limit from Nature Communications.
- We split Figure 4 into 2 Figures (Figure 4 and Figure 5) as suggested by the Editor.
- We moved the part of the text, described the morphology of the globular MIPs into the Figure Caption as suggested by the Editor.

REVIEWERS' COMMENTS:

Reviewer #1 (Remarks to the Author):

The authors provided newly updated data processed without binning which showed a significant improvement on data quality, and made it particularly convincing about the seam location. The authors have also provided updated Fourier shell correlation (FSCs). Most of the FSC profiles look reasonable except for the PF one, which shows significant raise-ups at the high frequency regions. It indicates this FSC profile would not be reliable for a resolution estimation. Therefore, the authors may want to be careful with the claim of the 4.2Å resolution based on this FSC. I would like

to suggest to revisit the FP FSC calculation. Other than that, the revised manuscript represents a significant improvement, and I believe it can be published.

We revisited the FSC calculation. Here is the soft-edge mask that we used for the resolution measurement (in YZ (top left), XY (bottom left) and XZ (bottom right))

The resolution measurement using EMDDB FSC Validation server. It reports 4.3 Angstrom without any bump at frequency ~ 0.30 ($1/\text{\AA}$).

The resolution measurement using corrected FSC measurement by Relion yields 4.6 \AA (Black line).

As you can see, the bump in the corrected FSC at frequency ~ 0.30 ($1/\text{\AA}$) is still there but much less pronounced than in the previous version of the manuscript. We took the conservative approach and use 4.6 \AA . We updated Supplementary Figure 1 accordingly.

There are also some minor errors in the manuscript as following:

Page 4, paragraph 1, line 3: “... fitted well in doublet map...” should be “...fitted well in the doublet map...”.

Page 4, paragraph 3, line 1: “....local curvature of doublet lattice...” should be “...local curvature of the doublet lattice...”.

Page 5, paragraph 3, line 8: “....showing that doublet can be formed without the inner junction structure” should be “...showing that doublets can be formed without the inner junction structure ...”.

Page 6, last paragraph, line 4: “....also show insertions into tubulin lattice...” should be “...also show insertions into the tubulin lattice ...”.

Page 7, paragraph 6, 1st line: “MIP6 appears to have in fact 16-nm repeating unit...” should be “MIP6 appears to have in fact a 16-nm repeating unit...”.

Page 7, last paragraph, 1st line: “We also observed 16-nm repeating unit density...” should be “We also observed a 16-nm repeating unit density...”.

Page 8, 2nd paragraph, line 5: “...two tubulin dimers and between α -...” should be “...two tubulin dimers and between α -...” i.e. : needs a space between the “between α -“

Page 9, paragraph 3, line 4: “...with an earlier study showed that...” should be “...with an earlier study which showed that...”

Page 12, 1st paragraph, line 3: “...difference biophysical effects...” should be “...different biophysical effects...” : i.e., change “difference” to “different”.

Page 18, paragraph 3, line 8: “The raw data were converted into...” should be “The raw data was converted into...”.

We fixed all of them.

Reviewer #2 (Remarks to the Author):

The revised manuscript is much improved in presentation of the structural details and the FSC for resolution. Only one remaining concern is that it is still difficult to see if α -tubulin and β -tubulin are well distinguished, especially at the seam between A9 and A10 shown in the right panel of Fig. 1c. What I would do to make it clearer is to present only the density map in the right panel of Fig. 1c without the fitted model and indicate a few density features with which readers can easily distinguish α -tubulin from β -tubulin.

It is indeed difficult to distinguish α -tubulin and β -tubulins from just looking from a fixed viewpoint at the surface of the map at 5.7 Angstrom resolution. Therefore, we don't think that simply displaying the map without the fitted model would help. We did rigorous analysis as described in the Material & Methods and illustrated in Supplementary Fig. 1e in order to make sure the assignment is correct. Also, interested readers can download the map and fitted model from EMDB to verify.

Also, Supplementary Fig. 1e should be magnified make it easy for readers to see the differences between α -tubulin and β -tubulin.

We updated the Supplementary Figure 1 with bigger panel for e.

Reviewer #3 (Remarks to the Author):

Ichikawa et al.: High-resolution structure of the doublet microtubule:

This paper shows cryo-EM reconstructions of microtubule doublets and thereby identifies a large number of so-called microtubule inner proteins (MIPs) who have been seen before, but no such detailed analysis has been reported yet. The authors exploit the newest technologies on the hardware (e.g. direct electron detectors) and the software sides, and present a stunning piece of work.

I was already very positive about this work the first time I saw it. The paper certainly improved even more due to the reviewer's comments. I almost wonder if this work would not give sufficient material for two publications ... As the first version, this is still an extremely overloaded paper with huge amounts of supplementary data, but if the editors are fine with this format I do not see any reason to delay publication. This is an important piece of data for a very active field of research, and even if some claims (e.g. the overall resolution of 0.62 nm at the very challenging 3sigma cutoff) are a bit too optimistic, especially for the MIP structures, this is a solid work. Useful resolution improves in steps, and often a range from 6-8Å is not really showing anything new but only sharpen edges. E.g. to really see the pitch of α -helices (as one reviewer pointed out) better resolution will be necessary. High resolution claims of MIP structures are not adding much to the data and may be corrupted by averaging artefacts, but the important part is that the authors were able to resolve and identify all the different MIPs presented. The addition of density images greatly helped to make the story more believable, at least for old-school microscopists as myself. I recommend this work for publication, provided the data has been properly uploaded to a 3-D EM database, and the editors have no problem with the cramped data presentation.

We have deposited all the maps and corresponding model.

The headers for the 16-nm and 48-nm maps have been released in EMDB. The 8-nm map and the PF-map and the corresponding PDB have just been approved and will be released in the next release cycle of EMDB.

Link of already released headers.

<http://www.ebi.ac.uk/pdbe/entry/emdb/EMD-8532>

<http://www.ebi.ac.uk/pdbe/entry/emdb/EMD-8537>